# Perception of root-active CLE peptides requires CORYNE function in the phloem vasculature

Ora Hazak[1,†], Benjamin Brandt[2,†], Pietro Cattaneo[1], Julia Santiago[2], Antia Rodriguez-Villalon[1], Michael Hothorn[2,*] (iD) & Christian S Hardtke[1,**] (iD)

## Abstract

*Arabidopsis* root development is orchestrated by signaling pathways that consist of different CLAVATA3/EMBRYO SURROUNDING REGION (CLE) peptide ligands and their cognate CLAVATA (CLV) and BARELY ANY MERISTEM (BAM) receptors. How and where different CLE peptides trigger specific morphological or physiological changes in the root is poorly understood. Here, we report that the receptor-like protein CLAVATA 2 (CLV2) and the pseudokinase CORYNE (CRN) are necessary to fully sense root-active CLE peptides. We uncover BAM3 as the CLE45 receptor in the root and biochemically map its peptide binding surface. In contrast to other plant peptide receptors, we found no evidence that SOMATIC EMBRYOGENESIS RECEPTOR KINASE (SERK) proteins act as co-receptor kinases in CLE45 perception. CRN stabilizes BAM3 expression and thus is required for BAM3-mediated CLE45 signaling. Moreover, protophloem-specific *CRN* expression complements resistance of the *crn* mutant to root-active CLE peptides, suggesting that protophloem is their principal site of action. Our work defines a genetic framework for dissecting CLE peptide signaling and CLV/BAM receptor activation in the root.

**Keywords** CLAVATA; CLE45; MAKR5; receptor kinase; SERK

**Subject Categories** Development & Differentiation; Plant Biology; Signal Transduction

## Introduction

Receptor kinases (RKs) are key regulators of growth and development in higher plants such as the model organism *Arabidopsis thaliana* (*Arabidopsis*). There are ~180 *Arabidopsis* RKs with extracellular leucine-rich repeat (LRR) domains, many of which can perceive peptide ligands, including members of the CLAVATA3/ EMBRYO SURROUNDING REGION (CLE) peptide family [1,2]. CLE peptides are encoded endogenously and translated as prepropeptides, which are secreted and processed to yield mature 12–13 amino acid, bioactive peptides [2–4]. In some cases, their activity is amplified by post-translational modifications, such as proline hydroxylation and additional arabinosylation [5,6]. *Arabidopsis* contains 32 *CLE* genes, some of which encode redundant peptides, thereby giving rise to 27 distinct CLE peptides [3,7,8]. Several of these peptides have been shown to play roles in root development, and chemically synthesized versions of many of them suppress *Arabidopsis* root growth in tissue culture when applied at nM to low μM concentrations (called root-active CLEs in the following) [9–11]. However, the perception mechanism for most CLE peptides in the root, including their receptors and co-receptors, remains unknown to date.

Genetic and biochemical studies have identified several LRR-RKs involved in the perception of individual CLE peptides. The outstanding, classic example is CLAVATA 1 (CLV1), which directly binds the CLV3 peptide to regulate stem cell homeostasis in the shoot apical meristem [4,12–17]. The CLV1-related LRR-RK BARELY ANY MERISTEM 3 (BAM3) is required to mediate the suppression of protophloem sieve element differentiation in the root meristem by CLE45 application [9,18,19]. PHLOEM INTERCALATED WITH XYLEM (PXY; a.k.a. TDIF RECEPTOR [TDR]) senses the redundant CLE41/44 peptides (a.k.a. TRACHEARY ELEMENT DIFFERENTIATION INHIBITORY FACTOR [TDIF]) to regulate vascular stem cell proliferation in secondary growth [20–25].

High-affinity ligand sensing and receptor activation of plant LRR-RKs relies on their interaction with shape-complementary co-receptor kinases [1,26–28]. For instance, the LRR-RK BRASSINOSTEROID INSENSITIVE 1 (BRI1) employs the SOMATIC EMBRYOGENESIS RECEPTOR KINASE (SERK) family co-receptors SERK1 and SERK3 to transmit the signal triggered by the small molecule ligand brassinosteroid [29]. The LRR-RK HAESA also relies on SERK proteins to transduce the signal triggered by the peptide ligand IDA, which is related to CLE peptides in structure [30,31]. Consistently, it was recently suggested that SERK1 also plays a role in PXY-mediated CLE41/44 signal transduction [25]. Thus, *Arabidopsis* SERKs have been implicated in multiple signaling pathways, comprising CLE as well as other peptide signals, hormonal cues, and pathogen-derived ligands [32,33]. Beyond PXY, however, it

1   Department of Plant Molecular Biology, University of Lausanne, Lausanne, Switzerland
2   Structural Plant Biology Laboratory, Department of Botany and Plant Biology, University of Geneva, Geneva, Switzerland
    *Corresponding author. Tel: +41 223793013; E-mail: michael.hothorn@unige.ch
    **Corresponding author. Tel: +41 216924251; E-mail: christian.hardtke@unil.ch
    †These authors contributed equally to this work

remains unclear to what degree SERKs could be involved in the perception of CLE peptides.

For CLV1, different types of potential, context-dependent co-receptors have been described. In one scenario, CLV3 is perceived in association with CLV2, a receptor-like protein (RLP) that is comprised of extracellular LRRs and a transmembrane domain but lacks a kinase domain [12,34]. CLV2 in turn dimerizes with CORYNE (CRN), which consists of a transmembrane domain and an intracellular pseudo-kinase domain [35,36]. Other findings point to a CLV1-independent role of CLV2-CRN in CLV3 perception, possibly in conjunction with the CLV1-related LRR-RKs BAM1 and BAM2 [35,37,38]. Moreover, it was found that CLV2-CRN is required for the perception of many if not all root-active CLE peptides [11,35,39,40]. Finally, CLV1 has been implicated in stem cell homeostasis in the root meristem, where it presumably perceives CLE40 together with the non-LRR-RK ARABIDOPSIS CRINKLY 4 (ACR4) [41,42]. Likewise, BAM1 and RECEPTOR-LIKE PROTEIN KINASE 2 are also thought to play a role in CLE perception in the root [43]. In this study, we show that BAM3 is a *bona fide* CLE45 receptor, which appears to operate independent of SERK proteins. Moreover, we demonstrate that phloem-specific CRN expression is not only required for perception of CLE45, but of all root-active CLE peptides tested in this study, possibly by stabilizing expression of their receptors.

# Results

### BAM3 is a CLE45 receptor

Originally, we isolated *bam3* as a second-site suppressor of loss of function in *BREVIS RADIX (BRX)*, which encodes a positive regulator of root protophloem sieve element differentiation, suggesting that *BRX* antagonizes the CLE45-BAM3 pathway [9,19]. Root protophloem differentiation (and thereby root growth) of *bam3* loss-of-function mutants is not impaired by exogenously applied CLE45 levels that suppress this process in wild-type plants, suggesting that BAM3 could act as a CLE45 receptor [9]. Subsequent results strengthened this notion [19,44] and also ruled out proposed alternative CLE45 receptors in the root [18]. However, direct evidence for CLE45-BAM3 interaction is still missing. We isolated additional CLE45-insensitive *bam3* loss-of-function alleles from our genetic screen [9] (Fig EV1A), including non-synonymous mutations leading to amino acid changes in the ligand-binding LRR and the cytoplasmic kinase domains (Fig 1A). Mutation of threonine 150 to isoleucine presumably interferes with proper folding of the BAM3 LRR domain, as does mutation of glycine 364 to arginine. Serine 303, however, maps to the inner face of the BAM3 LRR domain and is located in a surface region, which forms the peptide binding sites in the structurally related IDA receptor HAESA [30] and in the CLE41/44 receptor PXY [25] (Fig 1A and B). Two missense mutations in the BAM3 kinase domain (P883S, G901E) map to the core of the kinase C-lobe and may interfere with the proper folding or activity of the BAM3 kinase module. Together, our genetic analysis suggests that both the extracellular and cytoplasmic portions of BAM3 are important for the function of the receptor.

We produced the BAM3 LRR domain by secreted expression in insect cells and tested if the purified ectodomain interacts with a synthetic CLE45 peptide in isothermal titration calorimetry (ITC) assays. We found that BAM3 bound CLE45 with a $K_d$ of ~120 nM and with 1:1 stoichiometry (Fig 1C). The binding affinity for CLE45 to BAM3 was about 10-fold lower than CLE41/44 binding to the LRR ectodomain of PXY (Fig EV1B) [25]. BAM3 showed specific CLE45 binding, as the sequence-related CLV3 peptide, which is not expressed in the root [40], bound with much lower affinity ($K_d$ ~10 μM) (Fig EV1C). Importantly, N-terminal extension of CLE45 or CLV3 by a tyrosine residue (initially used to quantify the peptide concentrations) rendered the engineered peptides non-bioactive and drastically reduced binding to the BAM3 ectodomain (Fig EV1D–F). Prompted by our recent finding that the peptide hormone IDA is structurally related to CLE peptides, we created a BAM3 homology model based on the HAESA-IDA complex structure [30] to predict the CLE45 binding surface in BAM3 (Fig 1B). In our homology model, BAM3 residues Q226, Y228, and Y231 from the LRR domain form a part of the CLE45 binding surface. Consistently, binding of CLE45 to a purified BAM3 ectodomain in which Q226, Y228, and Y231 were mutated to alanines (BAM3$^{QYY}$) was ~8 times weaker when compared to the wild-type LRR domain (Fig 1C).

To test the relevance of these mutations *in planta*, we re-created them in a full-length *BAM3* coding sequence to express the mutant protein as a CITRINE fusion (*BAM3::BAM3$^{QYY}$-CITRINE*). First, we checked the subcellular localization of BAM3$^{QYY}$-CITRINE and wild-type BAM3-CITRINE (*BAM3::BAM3-CITRINE*) in transient transformation of tobacco (*Nicotiana benthamiana*) leaf cells. In this system, both fusion proteins showed similar plasma membrane localization as well as some internal, likely endoplasmic reticulum signals (Fig EV2A). In *Arabidopsis*, BAM3$^{QYY}$-CITRINE was specifically expressed in the protophloem, with a similar profile of subcellular (plasma membrane and internal) localization and abundance as wild-type BAM3-CITRINE (Figs 1D and EV2B). However, unlike BAM3-CITRINE, BAM3$^{QYY}$-CITRINE was neither able to restore the *brx* phenotype when introduced into a *bam3 brx* double mutant (Fig 1E and F), nor able to restore CLE45 sensitivity when introduced into a *bam3* single mutant (Fig 1G and H). In summary, these data reinforce the view that BAM3 is the genuine CLE45 receptor in the context of the root protophloem [18].

### Individual SERKs are not necessary to perceive root-active CLE peptides

SERK proteins have recently been shown to act as co-receptors for the CLE41/44 receptor PXY [25,45] and for many other LRR-RKs [46]. In the case of the peptide receptor HAESA, complex formation with SERK1 allows for the specific and high-affinity sensing of IDA, and HAESA and SERK1 form stable, IDA-dependent heteromeric complexes *in vitro* [30]. To test whether SERK proteins could be involved in the sensing of other CLE peptides in the root, we surveyed the response of *serk* mutants to 14 root-active CLE peptides (CLV3, CLE8, CLE9/10, CLE11, CLE13, CLE14, CLE16, CLE18, CLE20, CLE21, CLE25, CLE26, CLE40, CLE45), which were selected for their significant, reproducible impact on root growth at 50 nM concentration. In general, the response of representative *serk* loss-of-function mutants (alleles *serk1-3, serk2-1, serk3-1, serk4-1,* and *serk5-1*) was largely similar to wild type (Fig 2A). Interestingly, another *serk1* allele, *serk1-1*, showed resistance to CLE45 application and was as insensitive as *bam3* (Fig EV2C). However, CLE45 resistance and the *serk1-1* mutation segregated freely in outcrosses

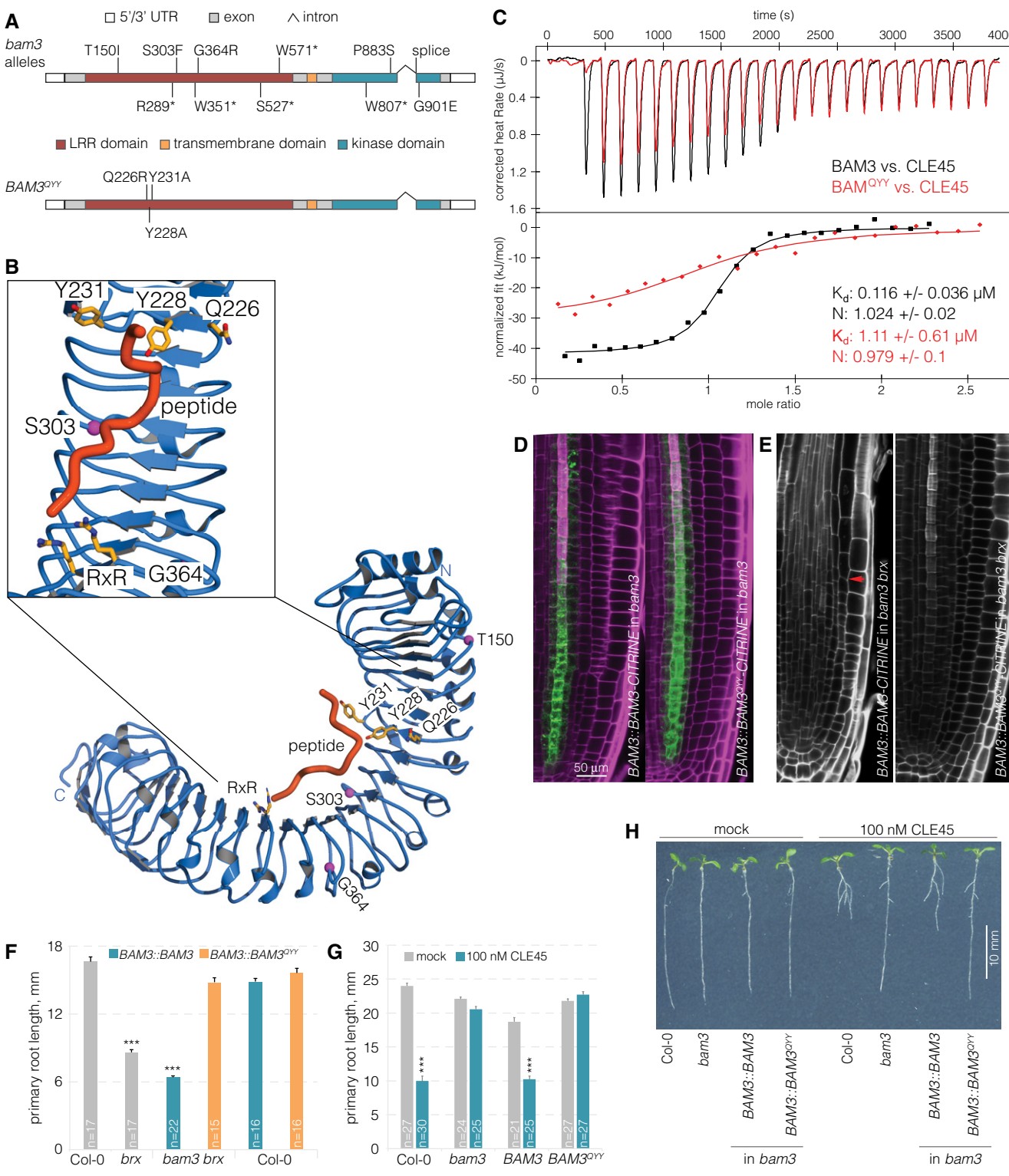

Figure 1.

to a *brx* mutant (i.e., only six out of 24 genotyped plants that were CLE45-resistant were also homozygous for *serk1-1*), suggesting that the CLE45 resistance resulted from an unlinked background mutation. Whole-genome sequencing of the *serk1-1* plants revealed a homozygous 28-bp deletion in *BAM3*, which would lead to a

frameshift after amino acid 699 and a premature stop codon six amino acids later, thereby deleting the kinase domain. Moreover, introduction of a transgenic *BAM3* copy restored CLE45 sensitivity of *serk1-1* (Fig EV2C). No complementation was observed with *SERK1* constructs that were reported to rescue the *serk1-1 serk2-1*

◄

**Figure 1.  BAM3 is a CLE45 receptor.**

A  Schematic overview of the *BAM3* gene structure. *bam3* loss-of-function mutations that were isolated as second-site suppressors of *brx* loss of function (top); amino acid point mutations predicted to disrupt BAM3-CLE45 interaction (bottom).

B  Ribbon diagram of a homology model of the BAM3 LRR ectodomain (in blue) based on the HAESA ectodomain (PDB-ID 5IXO). Magenta spheres indicate the position of genetic BAM3 missense mutations, and residues forming part of the predicted CLE45 binding site are shown in bonds representation (in yellow). The position of CLE45 has been inferred from an IDA-HAESA complex (PDB-ID 5IXQ).

C  Isothermal titration calorimetry (ITC) of purified BAM3 wild-type (black) or mutant BAM3$^{QYY}$ (red) extracellular domains vs. CLE45 peptide. N: stoichiometry, $K_d$ dissociation constant. Shown are experimental values $\pm$ fitting errors (95% confidence interval).

D  Expression of BAM3-CITRINE wild-type or mutant BAM3$^{QYY}$ fusion protein (green fluorescence) under control of the native *BAM3* promoter in root meristems of *bam3* mutants (magenta fluorescence: calcofluor white cell wall staining) (confocal microscopy).

E  Primary root meristems of *bam3 brx* double mutants carrying the indicated transgenes. Red arrow on the left panel indicates the approximate position of the final dividing cortex cell, which is out of range in the right panel.

F  Primary root length of 5-day-old seedlings of the indicated genotypes.

G  Primary root length of 7-day-old seedlings of the indicated genotypes in mock or CLE45 condition.

H  Representative 7-day-old seedlings of the indicated genotypes grown on standard mock or CLE45-containing media.

Data information: Differences as compared to Col-0 (F) or mock (G) are not statistically significant unless indicated (Student's *t*-test); \*\*\**P* < 0.001; mean $\pm$ s.e.m.

double-mutant shoot phenotype [47] (Fig EV2D). Therefore, the *serk1-1* line should be considered a *bam3 serk1* double mutant.

### SERK1 and SERK3 do not act as co-receptors in BAM3-mediated CLE45 signaling

In summary, none of the five *serk* mutants displayed CLE45 resistance. A signaling function of *SERK* genes in CLE45 perception might be masked by genetic redundancy, and therefore, the notion that SERK1 could be a BAM3 co-receptor still appeared attractive, especially given recently reported evidence that SERK1 cannot only interact with HAESA, but also with PXY, in a ligand-dependent manner [25,30]. However, although the BAM3 and SERK1 kinase domains were able to transphosphorylate each other in an *in vitro* kinase assay (Fig EV2E), neither SERK1 nor SERK3 formed CLE45-dependent complexes with BAM3 *in vitro* (Figs 2B and EV2F). In contrast, SERK1 formed CLE41/44-dependent heterodimers with PXY (Fig EV2G), corroborating earlier results [25,45]. Consistent with our gel filtration experiments, we could not detect binding of the SERK1-LRR domain to BAM3 in the presence of CLE45 in quantitative ITC assays (Fig 2C), while SERK1 bound to HAESA in the presence of IDA with low nanomolar affinity [30]. Finally, although *SERK1* has been reported to be expressed throughout the stele [48,49], closer inspection of a transgene driving expression of a SERK1-CITRINE fusion protein under control of the *SERK1* promoter (*SERK1::SERK1-CITRINE*) suggested that *SERK1* is not expressed in developing protophloem sieve elements (Fig EV2H). In summary, the results suggest that SERK1 (and based on our biochemical studies also SERK3) is not a co-receptor for BAM3-mediated CLE45 signaling.

### CLV2 and CRN are necessary for full perception of root-active CLE peptides

Since our experiments did not support a role for SERK proteins in BAM3 receptor activation and CLE45 signal transduction, we assessed the relative contribution of other known CLV/BAM signaling components, CLV2 and CRN [11,35,37,39,50]. Both *CLV2* and *CRN* are expressed throughout most root tissues including the vascular cylinder [51]. We corroborated these findings by creating transgenic lines in which CITRINE fusions of CLV2 or CRN were

expressed under control of their native promoters (*CLV2::CLV2-CITRINE* and *CRN::CRN-CITRINE*). *CLV2::CLV2-CITRINE* displayed expression mostly in the stele (Fig EV3A), and *CRN::CRN-CITRINE* was expressed in the same domain (Fig EV3B). However, while CLV2 appeared to be evenly expressed throughout the vascular cylinder of the root tip, CRN was apparently enriched in the phloem poles. To investigate *CRN* mutation in the same background as all other lines used in this study, we obtained a *crn* loss-of-function mutant in the Col-0 accession. In this CRISPR/Cas9-generated *crn* allele, a single nucleotide insertion in front of the 7$^{th}$ codon leads to a frameshift and three subsequent premature stop codons after amino acid 10 [52]. This *crn* mutant displayed complete insensitivity to CLE45 concentrations that strongly suppress protophloem differentiation and thus root growth in wild type (Fig 2A). A survey of other root-active CLE peptides revealed that this *crn* loss of function also conferred strong resistance to all of them (Fig 2A), corroborating the results for other *crn* alleles in different parental backgrounds [11]. As expected, we obtained very similar results when analyzing *clv2* loss-of-function mutants (Fig 2A). The *CLV2::CLV2-CITRINE* and *CRN::CRN-CITRINE* constructs complemented the respective mutants, indicating their functionality (Fig EV3C and D). Taken together, our experiments confirm that CLV2 and CRN are necessary to mediate full sensitivity to all root-active CLE peptides monitored in this study.

### The phloem is a crucial site of action for root-active CLE peptides

To test whether the CLE45 resistance of *crn* mutants reflects CRN activity in the developing protophloem, we expressed a transgenic CRN-CITRINE fusion under control of the *BAM3* promoter in *crn* mutants (Fig 3A–C). Interestingly, in these lines, we not only observed restored sensitivity to CLE45 (Fig 3D), but also to the similarly acting CLE26 [44] as well as other strongly root-active CLE peptides (Fig 3E). To monitor the CLE peptide effects in more detail, we transiently treated transgenic lines that expressed a nuclear localized fluorescent marker under control of the *COTYLEDON VASCULAR PATTERN 2* (*CVP2*) promoter (*CVP2::NLS-VENUS*) in wild-type background. *CVP2* is very specifically expressed in the developing sieve elements of the root meristem and is also a specific marker for their differentiation process [44]. Investigation of *CVP2::NLS-VENUS* seedlings after 24-h CLE peptide treatment indicated

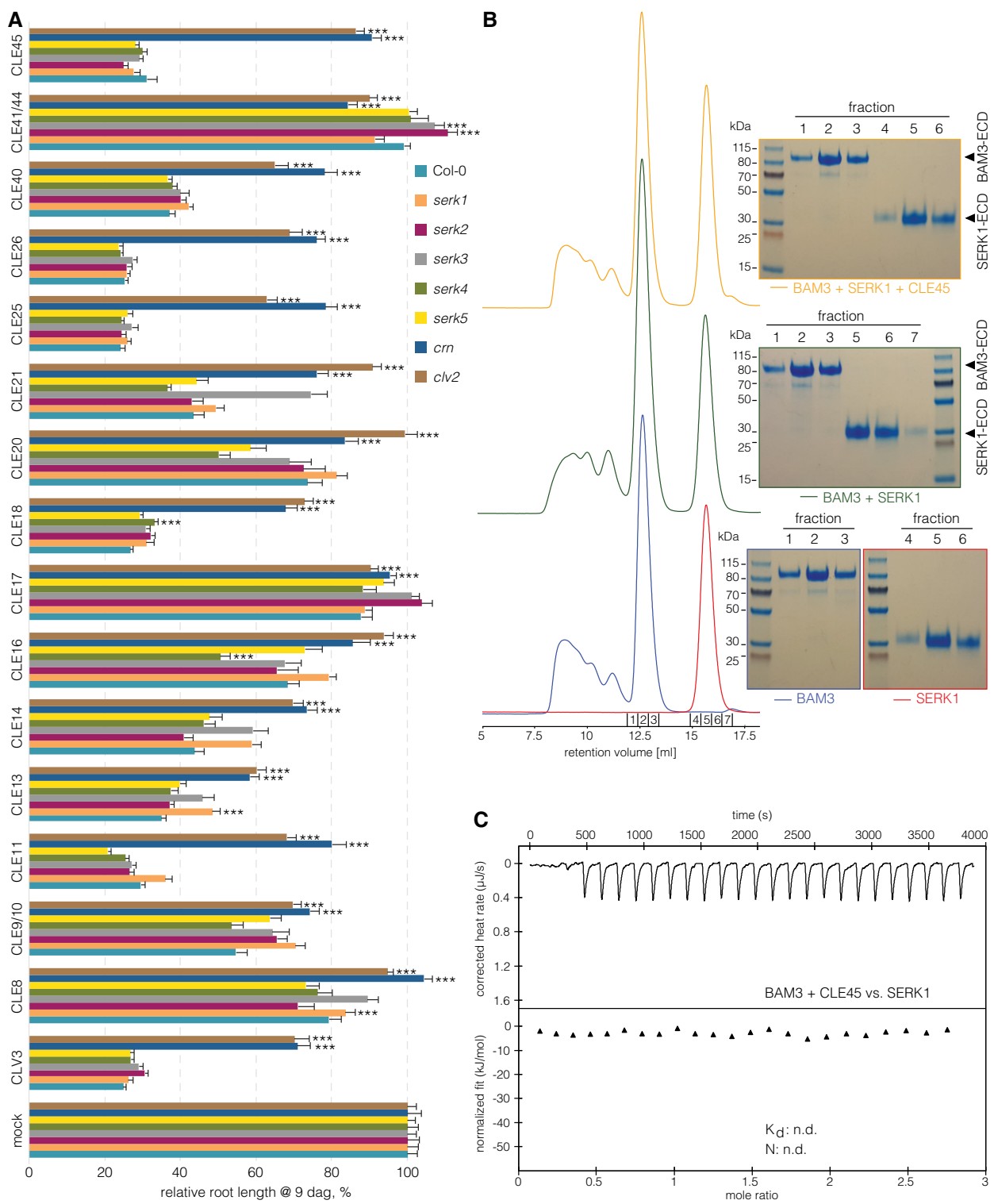

**Figure 2.  Root growth CLE peptide resistance of mutants in potential co-receptors of CLE signaling pathways.**

A  Primary root length of 9-day-old seedlings of indicated genotypes, grown on mock or 50 nM CLE peptide. Fourteen root-active peptides and two controls (CLE17 and CLE41/44) are shown. *n* ≥ 12 per column.

B  Analytical size-exclusion chromatography. The BAM3 and SERK1 LRR domains elute as monomers when run stand-alone (blue and red traces), in combination (green trace), and in the presence of CLE45 (yellow trace). SDS–PAGE gels of the corresponding peak fractions are shown alongside.

C  ITC of the purified BAM3 extracellular domain in the presence of CLE45 peptide titrated against the purified SERK1 extracellular domain. n.d.: not detectable.

Data information: Differences as compared to Col-0 (A) are not statistically significant unless indicated (Student's *t*-test); \*\*\**P* < 0.001; mean ± s.e.m.

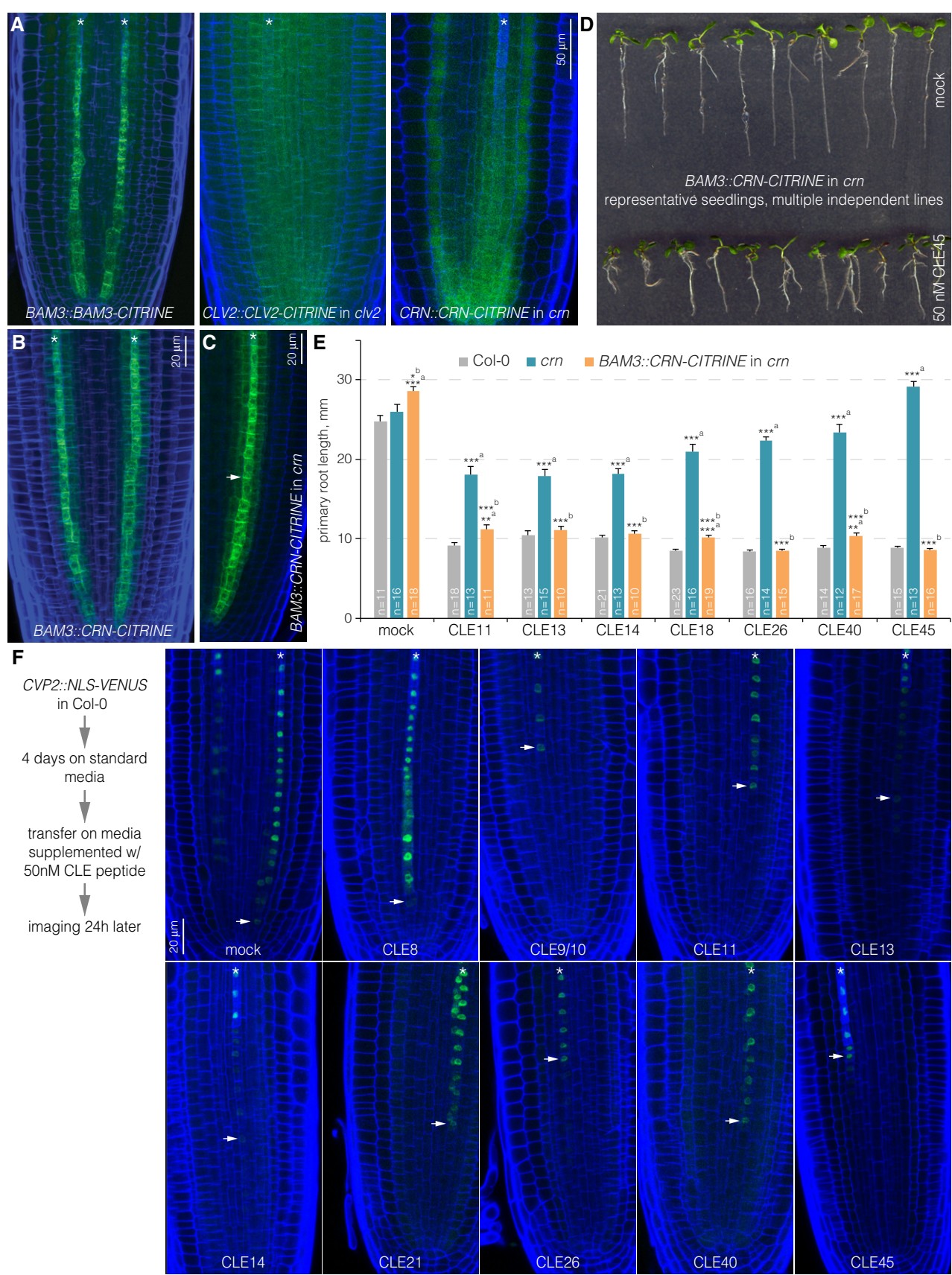

Figure 3.

**Figure 3.  Protophloem-specific CRN action in the root meristem.**

A    Expression pattern of BAM3, CLV2, and CRN-CITRINE fusion proteins (green fluorescence) under control of their native promoters (blue fluorescence: calcofluor white cell wall staining) (confocal microscopy). Asterisks mark developing protophloem sieve element strands.

B, C  Expression pattern of CRN-CITRINE fusion protein under control of the *BAM3* promoter in Col-0 (B) or *crn* (C) background. Arrowhead in (C) highlights plasma membrane-localized CRN.

D    Representative 7-day-old *crn* seedlings expressing CRN-CITRINE fusion protein under control of the *BAM3* promoter grown on mock or CLE45.

E    Primary root length of 7-day-old seedlings of indicated genotypes, grown on mock or 50 nM of selected CLE peptides. Differences as compared to [a]: Col-0 or [b]: *crn* are not statistically significant unless indicated (Student's *t*-test); *$P < 0.05$; **$P < 0.01$; ***$P < 0.001$; mean ± s.e.m.

F    Expression of nuclear localized, fluorescent VENUS protein under control of the *CVP2* promoter (which specifically marks developing protophloem sieve elements), 24 h after transfer from standard media to 50 nM of selected CLE peptides. Arrowheads indicate the cells closest to the tip in which *CVP2* expression was still detectable.

that indeed in practically all cases, protophloem development was perturbed to a large degree after CLE application (Fig 3F). While in some cases, a strong, immediate and specific suppression of protophloem differentiation was observed, in others the sieve element marker faded more gradually. However, in all cases, *CVP2* expression eventually disappeared toward the root tip. Together with the rescue of CLE peptide resistance through protophloem-specific *CRN* expression, these observations indicate that the developing phloem is a crucial site of action for root-active CLE peptides, even if some of them are not genuinely expressed in the root meristem and/or vasculature [53,54].

**CRN promotes CLE45 sensitivity by enhancing BAM3 expression**

To test whether CLV2-CRN might interact with BAM3, we next investigated these proteins in the cellular setting of the transient tobacco expression system. While BAM3 was mostly plasma membrane-localized (Fig EV4A), when expressed alone, CLV2 and CRN fusion proteins were mostly found inside cells and co-localized substantially with an endoplasmic reticulum marker (Figs 4A and B, and EV4B–D), in line with earlier findings [48,55]. Some plasma membrane localization could be observed at variable degrees in replicate experiments, which might be due to endogenous CLV2/CRN proteins, because as previously reported [55], co-expression of CLV2 and CRN resulted in increased delivery of both fusion proteins to the plasma membrane (Figs 4C and D, and EV4E and F). We confirmed these findings in the root vasculature of stable transgenic lines, by introducing the *BAM3::CRN-CITRINE* construct into the *clv2* mutant. While in wild type or the *crn* background CRN-CITRINE displayed substantial plasma membrane localization, it did not accumulate at the plasma membrane to the same extent in *clv2* mutants (Figs 4E and F, and EV3E). Conversely, CLV2-CITRINE fusion protein expressed under control of the *CLV2* promoter displayed some clear plasma membrane localization when expressed in the *clv2* mutant background, but mostly diffusive cytoplasmic localization when expressed in the *crn* mutant (Fig EV3F). Thus, as previously reported for the shoot, plasma membrane localization of CLV2 and CRN is largely interdependent in the root.

We next tested whether CLV2-CRN could interact with BAM3. We could not obtain the CLV2 ectodomain in sufficient quantity and quality for *in vitro* biochemical assays. In bimolecular fluorescence complementation (BiFC) experiments, we could not observe interaction of BAM3 with CRN, with the caveat that in this assay, CRN was supposedly not efficiently delivered to the plasma membrane. Also, in transient co-expression in tobacco, no co-localization was observed for CRN and BAM3 fusion proteins (Figs 4G and EV4G).

However, substantial co-localization occurred once a (non-fluorescent) HA-tagged CLV2 protein was co-expressed in addition (Figs 4H and EV4H and I). Co-localization of the three proteins was also observed when BiFC was performed for CLV2 and CRN in the presence of a fluorescent mTFP1-tagged BAM3 (Fig 4I). Moreover, a modest but robust BiFC interaction between BAM3 and CRN could be observed when (non-fluorescent) CLV2-HA was co-expressed as well (Fig 4J), as compared to a negative control (Fig 4K). Control experiments with BRI1 did not show such interaction (Fig 4L and M). Thus, it appears that in principle, BAM3 is capable of interaction with the CLV2-CRN dimer in a cellular setting. Such interaction could actually occur *in planta*, since both BAM3 and CRN displayed plasma membrane localization in developing sieve elements, which appeared to be mostly shootward for BAM3 (Fig 4N and O).

Despite the possibility for BAM3-CLV2/CRN interaction, it appears unlikely that CLV2-CRN acts in a capacity of co-receptor in CLE45 perception, since CRN is a pseudokinase [36]. To further investigate the role of *CRN* in CLE45 signaling, we thus conducted additional genetic experiments. Second-site loss of function in BAM3 or in its downstream signaling component MEMBRANE-ASSOCIATED KINASE REGULATOR 5 (MAKR5) suppresses *brx* mutant phenotypes [9,18]. Similarly, *clv2* or *crn* second-site mutation substantially rescued the protophloem differentiation, root meristem size and root growth defects of *brx* mutants (Fig 5A and B). The phenotype of *brx crn* double mutants qualified *crn* as a strong but partial *brx* suppressor, with overall rescue roughly comparable to *brx makr5* double mutants [18], but less penetrant than in *bam3 brx* double mutants [9]. Thus, *crn* loss of function dampens CLE45 signaling in the *brx* background, where the CLE45-BAM3 pathway is apparently hyperactive [9,18]. Another feature of CLE45 perception is the accumulation of MAKR5-GFP fusion protein in developing protophloem sieve elements upon CLE45 treatment [18]. This response, which appears to be triggered by post-translational events since it still takes place in the presence of cyclo-heximide (Fig 5C), is abolished in *bam3* mutants and was likewise undetectable in *crn* mutants (Fig 5D). Therefore, CRN is required for CLE45 signaling as judged by all established criteria. To investigate whether *crn* loss of function can affect the expression or subcellular localization of BAM3, we introduced the *BAM3::BAM3-CITRINE* construct into the *crn* mutant background. Indeed, we observed substantially reduced overall BAM3-CITRINE abundance in the *crn* mutant, especially at later stages of protophloem development (Fig 5E and F). Since *BAM3* gene expression was not affected in *crn* root tips (Fig 5G), this apparently reflected post-translational regulation. We verified this observation by crossing individual *BAM3::BAM3-CITRINE* lines in *crn* background with a *crn*

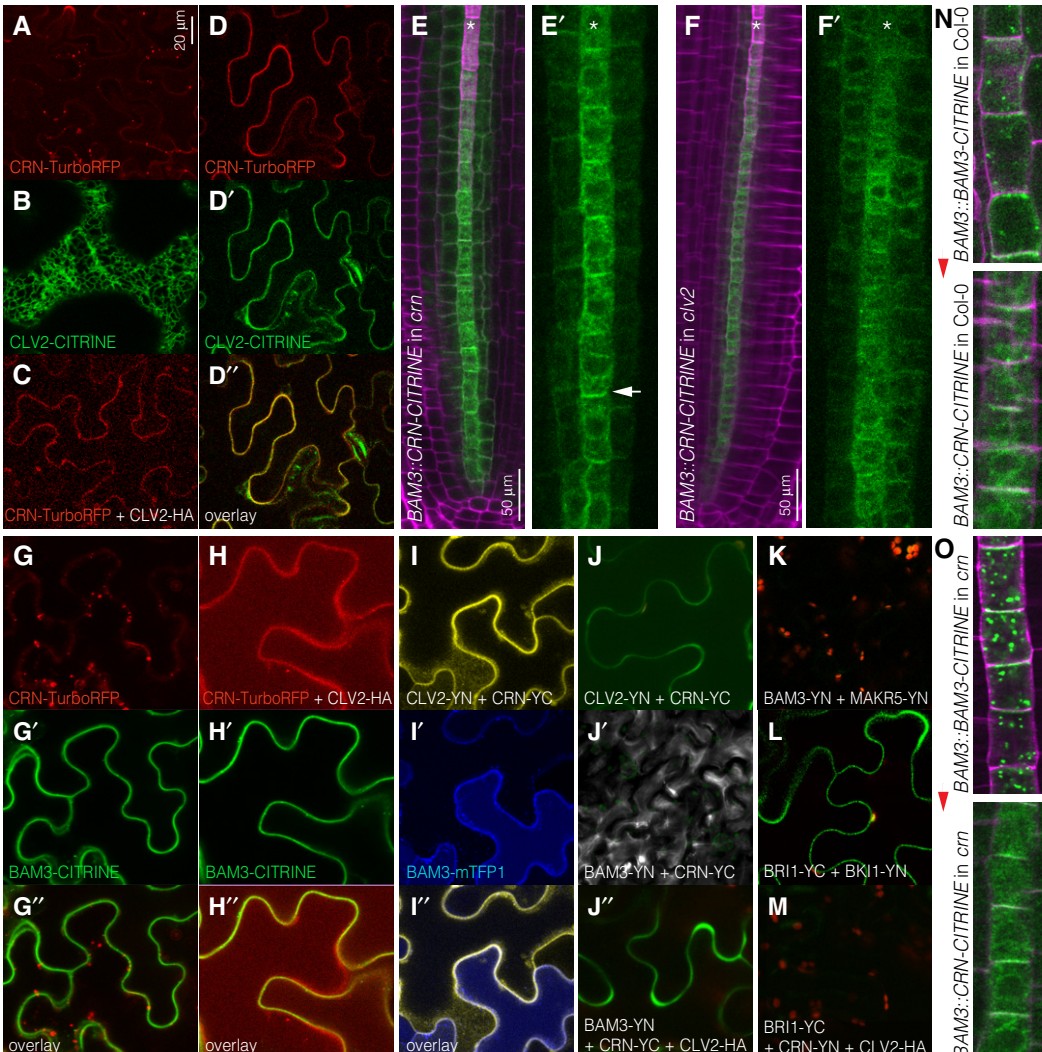

**Figure 4.  Co-localizations and interactions of BAM3, CRN, and CLV2.**

A    Transient expression of CRN-TurboRFP fusion protein (red fluorescence) in tobacco (*Nicotiana benthamiana*) leaf epidermal cells, under control of a constitutive promoter (confocal microscopy).

B    Transient expression of CLV2-CITRINE fusion protein (green fluorescence) in tobacco leaf epidermal cells.

C    Transient co-expression of CRN-TurboRFP fusion protein and (non-fluorescent) CLV2-HA fusion protein in tobacco leaf epidermal cells.

D    Transient co-expression of CRN-TurboRFP and CLV2-CITRINE fusion proteins in tobacco leaf epidermal cells. Red fluorescent and green fluorescent channels are shown separately and in overlay.

E, F    Expression pattern of CRN-CITRINE fusion protein (green fluorescence) under control of the *BAM3* promoter in *crn* (E) or *clv2* mutant (F), with corresponding close-ups (magenta fluorescence: calcofluor white cell wall staining). Asterisks mark developing protophloem sieve element strands. Arrowhead in (E′) highlights plasma membrane-localized CRN-CITRINE.

G    Transient co-expression of CRN-TurboRFP and BAM3-CITRINE fusion proteins in tobacco leaf epidermal cells. Red fluorescent and green fluorescent channels are shown separately and in overlay.

H    Transient co-expression of CRN-TurboRFP, BAM3-CITRINE, and (non-fluorescent) CLV2-HA fusion proteins in tobacco leaf epidermal cells. Red fluorescent and green fluorescent channels are shown separately and in overlay.

I    Bimolecular fluorescence complementation (BiFC) between transiently co-expressed CLV2 and CRN proteins fused to one half each of YFP (yellow fluorescence) in the presence of co-expressed BAM3-mTFP1 fusion protein (blue fluorescence). Yellow fluorescent and blue fluorescent channels are shown separately and in overlay.

J    BiFC between transiently co-expressed BAM3 and CRN proteins fused to one half each of YFP (green fluorescence) in the presence of co-expressed (non-fluorescent) CLV2-HA fusion protein in tobacco leaf epidermal cells (J″). Parallel control experiments for BiFC between CLV2 and CRN (J) and BAM3 and CRN (J′) are shown.

K    BiFC between transiently co-expressed BAM3 and MAKR5 proteins fused to one half each of YFP (red: chloroplast autofluorescence).

L    BiFC between transiently co-expressed BRI1 and BKI1 proteins fused to one half each of YFP (green fluorescence).

M    BiFC between transiently co-expressed BRI1 and CRN proteins fused to one half each of YFP in the presence of co-expressed (non-fluorescent) CLV2-HA fusion protein in tobacco leaf epidermal cells (red: chloroplast autofluorescence).

N, O    Expression of BAM3-CITRINE or CRN-CITRINE fusion proteins (green fluorescence) under control of the *BAM3* promoter in the developing sieve element cells close to the stem cells in Col-0 (N) or *crn* (O) background (magenta fluorescence: calcofluor white cell wall staining). Red arrowhead indicates rootward direction.

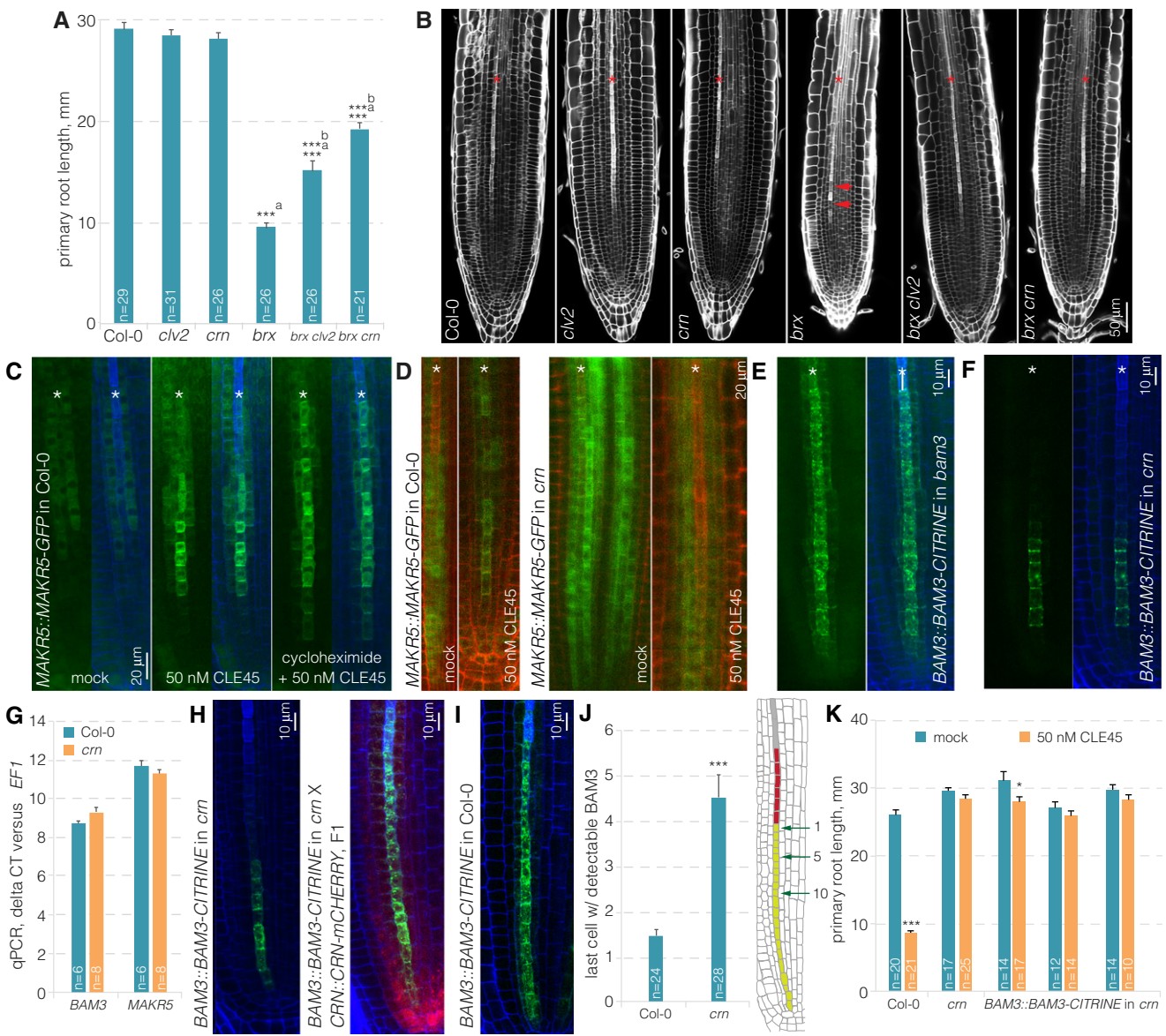

**Figure 5.   Genetic interactions between *CRN*, *BRX*, and *MAKR5*.**

A     Primary root length of 9-day-old seedlings of indicated genotypes.

B     Root meristems of indicated genotypes (white fluorescence: calcofluor white cell wall staining) (confocal microscopy). Red arrowheads indicate the "gap" cells in *brx*.

C     MAKR5-GFP fusion protein (green fluorescence) expressed under control of the native *MAKR5* promoter in response to CLE45 application in the presence or absence of cycloheximide (blue fluorescence: calcofluor white cell wall staining). Green channel is shown separately (left) and in overlay with blue channel (right).

D     Response of MAKR5-GFP fusion protein to CLE45 application in *crn* mutant background as compared to Col-0 control (red fluorescence: propidium iodide cell wall staining).

E, F  Expression of BAM3-CITRINE fusion protein under control of the *BAM3* promoter in developing protophloem of *bam3* (E) or *crn* (F).

G     qPCR of *BAM3* expression in Col-0 or *crn* root tips, with *MAKR5* as control, relative to the *EF1* housekeeping gene, representing the average for 2–3 technical replicates of three biological replicates, mean ± s.e.m. Differences were not statistically significant between Col-0 and *crn* (Student's *t*-test; *P* = 0.096 for *BAM3*, *P* = 0.273 for *MAKR5*).

H     Expression of BAM3-CITRINE fusion protein (green fluorescence) under control of the *BAM3* promoter in developing protophloem of *crn* (left panel) and in an F1 plant derived from a cross of the same line to a *crn* mutant complemented by a *CRN::CRN-mCHERRY* (red fluorescence) transgene (right panel).

I     Expression of BAM3-CITRINE fusion protein (green fluorescence) in Col-0, as a parallel control for panel (J).

J     Quantification of the last proliferating protophloem cell (light green cells in the root schematic) with detectable BAM3-CITRINE signal, with respect to the beginning of the protophloem transition zone (red cells).

K     Primary root length of 7-day-old seedlings of indicated genotypes on mock or CLE45 media, and several independent transgenic lines are shown.

Data information: Differences as compared to [a]: Col-0 or [b]: *brx* (A), mock (M), are not statistically significant unless indicated (Student's *t*-test); *\*P* < 0.05; \*\*\*P* < 0.001; mean ± s.e.m. Asterisks in (B–F) mark developing protophloem sieve element strands.

line that was complemented by a *CRN::CRN-mCHERRY* construct. In corresponding F1 plants that thus carried both hemizygous *BAM3:: BAM3-CITRINE* and *CRN::CRN-mCHERRY* transgenes in homozygous *crn* background, BAM3-CITRINE expression was restored to its wild-type levels (Fig 5H and I). This included expression throughout the developing sieve element cell file, which was one feature that could be easily quantified. While BAM3-CITRINE signal could practically always be observed up to the last cell before the protophloem transition zone in wild type, the signal was already completely absent in *crn* roots much earlier (Fig 5J). Finally, because increase in BAM3 dosage through *BAM3::BAM3-CITRINE* could not overcome the CLE45 resistance of *crn* (Fig 5K), it appears that CRN is essential for CLE45 perception because it is required for efficient BAM3 protein expression and its maintenance during sieve element development.

## Discussion

Plant LRR-RKs are central signaling hubs that can sense diverse ligands to control various facets of the plant life cycle, from plant–pathogen interactions to intrinsic developmental processes. Many of the currently known LRR-RKs require shape-complementary co-receptor kinases for receptor activation [46]. For instance, LRR-RKs of the SERK family can serve as co-receptors for BRI1 in brassinosteroid sensing [29], for FLAGELLIN INSENSITIVE 2 in innate immunity [56,57], for ERECTA and related RKs in stomata development [58], and for HAESA in abscission [30]. More recently, SERKs have also been implicated in CLE peptide sensing, as co-receptors of PXY as well as PXY-LIKE LRR-RKs [25,45]. In the case of the LRR-RK HAESA, the presence of SERK1 strongly increases the binding affinity for the IDA peptide (a ~60-fold increase from 20 μM to 350 nM) [30]. Consequently, while there is no detectable binding of the SERK1 ectodomain to HAESA in the absence of ligand, IDA-bound HAESA senses SERK1 with 75 nM affinity [30]. Structural comparison of the HAESA-IDA-SERK1 ternary complex with a CLE receptor complex containing PXY-CLE41/44-SERK2 revealed that both complexes are highly similar (root-mean-square deviation, r.m.s.d., is 2.3 Å comparing 782 $C_\alpha$ atoms) (Fig EV5A and B). However, in contrast to HAESA, the isolated PXY ectodomain binds CLE41/44 with nanomolar (10–30 nM), not micromolar, affinity (Fig EV1B) [25]. Interestingly, the SERK1 ectodomain binds relatively weakly to CLE41/44-bound PXY, suggesting that, despite their structural similarities, the activation mechanisms for HAESA and PXY may differ [30,45].

Our system-wide analysis of root-active CLE peptides in *Arabidopsis* suggests that only few CLE-sensing complexes may critically involve SERK co-receptor kinases or that CLE resistance phenotypes in *serk* mutants are caused by secondary effects. However, in our study, we did not investigate genetic redundancy between *SERK* genes. It appears possible that an array of higher order *serk* mutants will uncover fully redundant, overlapping roles of *SERKs* in CLE peptide sensing. Such analyses are substantially complicated by the dwarf and short root phenotypes of higher order *serk* mutants, however [59]. Moreover, despite their overall high sequence similarity, SERK proteins have diversified sufficiently to adopt potentially separate signaling roles [60]. Thus, their potential genetic requirement in CLE perception might be determined by a

combination of differential expression patterns and levels as well as protein structure variation.

Importantly, we could not find biochemical, genetic, or cell biological evidence that would support a role for SERKs in BAM3-mediated CLE45 sensing and signaling. While we could demonstrate CLE45 binding by BAM3 *in vitro* and observe matching *in planta* evidence, SERK1 or SERK3 did neither form CLE45-dependent or CLE45-independent complexes with BAM3, nor did *serk* mutants display CLE45-resistant phenotypes. Moreover, our finding that the *serk1-1* allele carries a *bam3* background mutation that confers CLE45 resistance emphasizes that full-scale analysis of *serk* mutant redundancy could only be considered reliable in conjunction with transgenic rescue. Reported phenotypes for *serk* multiple mutants that involve the *serk1-1* allele should thus be carefully considered in light of CLE45 resistance, especially with respect to vascular phenotypes [45]. Indeed, it appears possible that BAM3 also has a role in secondary growth [61].

Compared to *SERK* genes, the genetic requirement for *CLV2* and *CRN* in the full-scale sensing of all root-active CLE peptides that we investigated in this study was absolute. The results point to a generic rate-limiting role of CLV2-CRN in CLE receptor activity, for which multiple scenarios could be envisaged. In this respect, a classic role as co-receptor appears least likely, because CRN does not possess an active kinase domain and CLV2 is very different in size and sequence from SERK proteins [36]. Nevertheless, CLV2-CRN might participate as a component in receptor complexes, for instance to stabilize them or to recruit downstream components. Such a role would not be mutually exclusive with another possibility, a role of CLV2-CRN in promoting the plasma membrane delivery of LRR-RKs, or in enhancing their plasma membrane localization indirectly, for instance through molecular crowding. Our observations of reduced BAM3 abundance in *crn* mutants support the latter ideas, and it remains to be seen whether this will also apply to other, yet to be identified CLE peptide receptors in the root. So far, a conceptually similar role of CRN was not observed in the shoot [62]. However, such a function might be masked by the observed compensatory transcriptional cross-regulation between redundantly acting receptors [63], a scenario that apparently does not exist for BAM3 in the root [9,18].

Our most surprising finding is the observation that *CRN* activity in the developing protophloem was sufficient to restore sensitivity to all root-active CLE peptides investigated in this study. This observation is consistent with the more or less penetrant effect of all of these peptides on protophloem differentiation, irrespective of additional effects on root development that could be observed for individual CLEs. Thus, the results reinforce the emerging notion of the protophloem as a limiting, systemic organizer of overall root meristem development [9,19,64]. What remains enigmatic, however, is why *crn* mutants do not display an apparent morphological root phenotype if none of the root-efficient CLE peptides can be sensed properly? One possibility is that observations obtained from external CLE application might at least in part be misleading with regard to the genuine role of those peptides. Yet the CLV2/CRN module could function in certain conditions that upregulate CLE peptide levels and thereby might be crucial for root growth adaptation. Alternatively, it could also mean that most CLE peptide signaling is not essential for root development, at least in tissue culture conditions, or that compensatory, possibly non-peptide-mediated mechanisms

exist. It is important to note, however, that *crn* loss of function does not confer complete CLE peptide resistance, as exemplified by the strong, yet partial CLE45 resistance of *crn* as compared to *bam3*. Therefore, it appears possible that partially redundant CRN-related functions exist. Identification of additional genuine CLE peptide receptors in the root context might enable us to address this topic systematically in future studies.

In summary, we present a genetic framework for CLE peptide sensing in the *Arabidopsis* root that assigns distinct functions to various known receptor pathway components. Our data suggest that SERK proteins are involved in some, but not in the majority of the root CLE-sensing membrane signaling complexes, or alternatively act in a highly redundant manner. Instead, we provide evidence that CLV2 and CRN are part of root-active CLE peptide signaling pathways, possibly by controlling the expression, proper membrane localization, and/or stability of LRR-RK signaling complexes. Finally, our data suggest that the root protophloem is the crucial site of action of root-active CLE peptides. Nevertheless, they are apparently perceived by several distinct receptor complexes, many of which remain to be identified. The data presented in this study could serve as a resource to facilitate this task.

# Materials and Methods

Plant tissue culture, plant transformation, and common molecular biology procedures such as genomic DNA isolation, genotyping, sequencing, and peptide or inhibitor treatments were performed according to standard procedures as previously described [9,19,44,65].

### Sequence analysis of the *serk1-1* mutant

Whole-genome sequencing and data analysis of *serk1-1* mutants were performed as described [9]. The data and experimental details can be retrieved from the NCBI Short Read Archive at https://www.ncbi.nlm.nih.gov/Traces/study/?acc=SRP104341 [accessions: STUDY: PRJNA383544 (SRP104341); SAMPLE: serk1-1 (SRS2134599); EXPERIMENT: A_ (SRX2748469); RUN: serk1-1_R1_fused.fastq.gz (SRR5460456)].

### Plant materials, growth conditions, and physiological assays

All mutant and transgenic lines were in the *Arabidopsis* Columbia-0 (Col-0) background. The following previously described mutant alleles were used throughout: *brx-2, bam3-2, brx-2 bam3, clv2-1, serk1-1* and *serk1-3, serk2-1, serk3-1, serk4-1, serk5-1* [9,47,59,66–68]. The *crn* loss-of-function mutant allele (*crn-10*) carries a single nucleotide insertion in front of the 7th codon, which leads to a frameshift and three subsequent premature stop codons after amino acid 10 [52]. Observations with transgene constructs were confirmed in multiple independent transgenic lines.

### Transgenic constructs and lines

The *BAM3::BAM3-CITRINE, MAKR5::MAKR5-GFP,* and *CVP2::NLS-VENUS* constructs and transgenic lines have been described before [18,19,44]. The transgenic lines created for this study are summarized in Table EV1. All constructs used in this study were created with the multi-site GATEWAY cloning system according to standard protocols (Invitrogen). The constructs are listed in Table EV2. Oligonucleotides used in cloning procedures or for genotyping are listed in Table EV3.

### qRT–PCR

For qRT–PCR (qPCR), RNA was prepared from root tips of 7-day-old Col-0 and *crn* seedlings, and *BAM3* and *MAKR5* expressions were quantified relative to the *EF1* housekeeping gene on an Applied Biosystems Quantstudio 3 instrument as previously described [9,18].

### Transient expression and BiFC assays

For transient expression experiments, we used the 4th to 6th leaves of *N. benthamiana* plants. Infiltrations, co-localization, and BiFC analyses were essentially performed as previously described [69].

### Confocal imaging

Confocal images were obtained with Zeiss 700 or Zeiss 780 inverted confocal microscopes. All dual-color images were acquired by sequential line switching, allowing the separation of channels by both excitation and emission.

### Root counter staining

In some experiments, we used calcofluor white instead of standard propidium iodide (PI) staining for visualizing cell walls. To this end, 4- to 6-day-old seedlings were fixed in 4% paraformaldehyde in PBS for 45 min. After washing with PBS, seedlings were cleared with ClearSee solution [70] overnight. After incubation in 0.2% calcofluor white in ClearSee solution, seedlings were transferred to fresh ClearSee solution for 2–24 h before imaging.

### Peptides

Peptides were obtained at the 1–4 mg scale, > 75% pure (GenScript USA Inc.). The peptide sequences were as follows. CLV3: RTVPSGPDPLHH; CLE8: RRVPTGPNPLHH; CLE9/10: RLVPSGPN-PLHN; CLE11: RVVPSGPNPLHH; CLE13: RLVPSGPNPLHH; CLE14: RLVPKGPNPLHN; CLE16: RLVHTGPNPLHN; CLE17: RVVHTGPN-PLHN; CLE18: RQIPTGPDPLHN; CLE20: RKVKTGSNPLHN; CLE21: RSIPTGPNPLHN; CLE25: RKVPNGPDPIHN; CLE26: RKVPRGP-DPIHN; CLE40: RQVPTGSDPLHH; CLE41/44: HEVPSGPNPISN; CLE45: RRVRRGSDPIHN. N-terminally tyrosine-modified CLV3, CLE41/44, and CLE45 peptides were used as standards for quantification. Peptides for biochemical assays were synthesized by Peptide Specialty Laboratories, Heidelberg, Germany (see below).

### Protein homology modeling

Structural homologs of the BAM3 LRR (residues 30–651) and kinase (residues 699–903) domains were identified using the program HHpred [71], and homology modeling was done in Modeller [72] using the isolated LRR domain of HAESA (PDB-ID 5XIO) [30] and the kinase domain of BRI1 (PDB-ID 5LPB) [73] as templates.

## Protein expression and isolation of LRR-RLK ectodomains

The coding sequences for BAM3 (amino acids 30–651), PXY/TDR (30–647), SERK1 (24–213), and SERK3 (1–220) ectodomains were amplified out of *Arabidopsis* Col-0 cDNA using the PfuX7 polymerase (Norholm). The point mutations Q226A, Y228A, and Y231A for the BAM3$^{QYY}$-ECD were introduced by site-directed mutagenesis. All DNA fragments were cloned into a modified pFastBac1 vector (Geneva Biotech), fusing BAM3-, PXY-, and SERK1-ECD coding sequences with an N-terminal azurocidin signal peptide. To all ectodomain sequences, a C-terminal StrepII-9xHIS tandem affinity purification tag was added, and all constructs were confirmed by sequencing. Bacmids were generated by transforming the plasmids into *Escherichia coli* DH10MultiBac (Geneva Biotech), isolated, transformed into *Spodoptera frugiperda* Sf9 with Profectin transfection reagent (AB Vector) followed by viral amplification. Secreted protein was expressed by infecting *Trichoplusia ni* Tnao38 cells or for PXY-ECD *S. frugiperda* Sf9 with a viral multiplicity of 1, and incubation for 3 days post-infection. Cells were separated by centrifugation at 5,000 × *g* for 30 min, and the supernatant was filtered through 0.45 μm filters. The proteins were isolated from the medium by Ni$^{2+}$ (HisTrap excel; GE Healthcare) and subsequent StrepII (Strep-Tactin Superflow high capacity, IBA) affinity chromatography and further purified by size-exclusion chromatography (Superdex Increase 200) with 20 mM citrate pH 5 and 150 mM NaCl for BAM3 and SERK1/3 as buffer. For PXY and also one preparation of SERK1, the gel filtration was carried out with 10 mM Bis–Tris and 100 mM NaCl [45]. Molecular weights of all purified proteins were determined by MALDI-TOF mass spectroscopy (BAM3-ECD: 94,586 Da; PXY-ECD: 92,519 Da; SERK1-ECD: 27,551 Da; SERK3-ECD: 29,951 Da), and concentrations were measured via the absorption at 280 nm (corrected with the extinction coefficient for each protein).

## Gel filtration experiments

For each gel filtration experiment, 100 μg of SERK1-ECD or SERK3-ECD and equimolar amounts of either BAM3-ECD or PXY-ECD as well as 25 μM of CLE45 (RRVRRGSDPIHN), and 25 μM CLE41/44 (HEV-Hyp-SG-Hyp-NPISN) were used in the combinations indicated in the figures, incubated for 10 min after mixing and then subjected to gel filtration on a Superdex Increase 200 equilibrated with 20 mM citrate pH 5 and 150 mM NaCl, except for the gel filtration with PXY + SERK1 + CLE41/44, for which the column was equilibrated in 10 mM Bis–Tris and 100 mM NaCl [45]. The concentration of peptides modified N-terminally with a tyrosine (Y-peptides) was determined by their absorbance at 280 nm (corrected by the extinction coefficient). The concentrations for the peptides without tyrosine modification were determined by a quantitative colorimetric peptide assay (Pierce) using the respective Y-peptides as standards. The elution of proteins from the gel filtration column was monitored by absorption at 280 nm. Fractions indicated in the figures were separated on Bis–Tris polyacrylamide gels.

## Isothermal titration calorimetry

Isothermal titration calorimetry assays were run on a Nano ITC (1 ml standard cell; 250 μl syringe; TA Instruments) at 25°C in 20 mM sodium citrate pH 5 and 150 mM NaCl for all ITCs with BAM3 and 10 mM Bis–Tris pH 6 and 100 mM NaCl for the ITC with PXY. CLE41/44 (HEVPSGPNPISN), CLE45 (RRVRRGSDPIHN), CLV3 (RTV-HYP-SG-HYP-DPLHHH), and Y-CLV3 (YRTV-HYP-SG-HYP-DPLHH) were dissolved in the respective ITC buffer (20 mM sodium citrate pH 5 and 150 mM NaCl for CLE45, Y-CLE45, CLV3, and Y-CLV3; 10 mM Bis–Tris pH 6 and 100 mM NaCl for CLE41/44), and the following concentrations were used in the assays: BAM3-ECD vs. CLE45: 10 and 80 μM; BAM3-ECD vs. Y-CLE45: 9 and 154.5 μM; BAM3$^{QYY}$ vs. CLE45: 8.6 and 80 μM; BAM3-ECD vs. CLV3: 10 and 400 μM; BAM3-ECD vs. Y-CLV3: 8.6 and 175 μM; PXY-ECD vs. CLE41/44: 7.5 and 75 μM; BAM3-ECD + CLE45 vs. SERK1-ECD: 8.2 + 25 μM and 82 μM, respectively. For each experiment, 10 μl was repetitively injected into the cell in 150 s time intervals. The measured heat rates for the BAM3-ECD vs. CLE45, BAM3$^{QYY}$ vs. CLE45, BAM3-ECD vs. Y-CLE45, BAM3-ECD vs. CLV3, BAM3-ECD vs. Y-CLV3, and PXY-ECD vs. CLE41/44 were corrected by subtracting the heat rates measured for injecting CLE45, Y-CLE45, CLV3, Y-CLV3, or CLE41/44 into the ITC buffer, respectively. The data for the BAM3-ECD + CLE45 vs. SERK1-ECD measurement was corrected by subtracting heat rates acquired by injecting SERK1-ECD into a cell containing CLE45. Data analyses and modeling were carried out using the software supplied by the manufacturer (NanoAnalyze, version 3.5).

## Kinase domain protein expression and isolation as well as *in vitro* kinase assays

Kinase domain protein production and *in vitro* kinase assays were carried out as previously described [30]. In brief, the coding sequence for the cytosolic part of BAM3 (679–992; BAM3-KD) was amplified from *Arabidopsis* Col-0 cDNA. The DNA for the cytosolic region of SERK1 (264–625; SERK1-KD) was synthesized and codon-optimized for expression in *E. coli*. Both were inserted into an expression vector based on pET (Novagen) that gives rise to an N-terminal tag consisting of 8xHis-StrepII-Thioredoxin, which can be cleaved by a TEV-protease. Per site-directed mutagenesis point mutations were introduced into both coding sequences, resulting in inactive kinase domains for SERK1-KD (D447 to N; mSERK1-KD) and BAM3-KD (D854 to N; mBAM3-KD) [73]. All constructs were confirmed by sequencing and transformed into *E. coli* Rosetta 2 (Novagen). The bacteria were grown to an OD$_{A600}$ = 0.6, and protein expression was induced by adding IPTG to a final concentration of 0.5 mM followed by incubation for 18 h at 16°C. Cells were harvested by centrifugation at 5,000 × *g* and 4°C for 15 min, resuspended in buffer A (20 mM Tris–HCl pH 8, 500 mM NaCl, 4 mM MgCl$_2$, and 2 mM β-mercaptoethanol) to which 15 mM imidazole and 0.1% (v/v) Igepal were added, and lysed by sonication. The sample was then centrifuged at 35,000 × *g* and 4°C for 30 min, and the supernatant was used for a Co$^{2+}$ affinity purification. Co$^{2+}$ resin (HIS-Select Cobalt Affinity Gel, Sigma) was incubated with the cell lysate for 60 min at 4°C, subjected to a gravity flow column (Pierce), and washed twice with buffer A (+15 mM imidazole). Recombinant proteins were eluted in buffer A (+250 mM imidazole) and dialyzed in buffer B (20 mM Tris–HCl pH 8, 250 mM NaCl, 4 mM MgCl$_2$, and 0.5 mM TCEP). To BAM3-KD and BAM3$^{QYY}$-KD, TEV-protease was added before dialyses and then removed after dialyses by a second Co$^{2+}$ HIS-affinity purification (during this step, the cut tag was also removed). The proteins were then subjected to

gel filtration on a Superdex Increase 200 column using buffer B, collected, and concentrated in Amicon Ultra devices (10,000 MWCO cutoff).

To perform *in vitro* kinase assays, 1 µg of each kinase domain, in the combination indicated in the figure, was added in a total volume of 20 µl (buffer B). To start the reaction, 5 µCi of [γ-$^{32}$P]-ATP (Perkin-Elmer) was added and the reaction was carried out for 45 min at room temperature before being terminated by the addition of 4 µl of 6× SDS-loading dye and immediate incubation at 95°C for 7 min. SDS–PAGE in 4–15% gradient gels (TGX, Bio-Rad) separated the proteins, and the gels were subsequently stained with Instant Blue (Expedeon). Gel pictures were taken, and subsequently, an X-ray film was exposed to the gel in order to detect the radioactive signals of $^{32}$P.

Expanded View for this article is available online.

## Acknowledgements
We would like to thank Dr. Zachary Nimchuk for the *crn-10* mutant allele and Drs. N. Geldner and S. Yalovsky for vector plasmids. This work was supported by Swiss National Science Foundation grants 31003A_166394 (awarded to C.S.H.) and 31003A_156920 (awarded to M.H.), German Research Foundation (DFG) grant CRC 1101-D01 (awarded to M.H.), a Human Frontier Science Program Career Development Award (M.H.), and the European Molecular Biology Organization (EMBO) Young Investigator program (M.H.). A.R.V., J.S., and B.B. were supported by EMBO long-term post-doctoral fellowships.

## Author contributions
OH, BB, MH, and CSH designed the study and wrote the paper together. OH, BB, PC, JS, and AR-V performed experiments.

## Conflict of interest
The authors declare that they have no conflict of interest.

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
