## [Review Process File · EMBO Reports]

Manuscript EMBO-2016-43535

Perception of root-active CLE peptides requires CORYNE function in the phloem vasculature

Ora Hazak, Benjamin Brandt, Pietro Cattaneo, Julia Santiago, Antia Rodriguez-Villalon, Michael Hothorn, and Christian S. Hardtke

Corresponding authors: Christian Hardtke, University of Lausanne; Michael Hothorn, University of Geneva

Review timeline:

Submission date:	19 October 2016
Editorial Decision:	17 November 2016
Revision received:	14 March 2017
Editorial Decision:	19 April 2017
Revision received:	21 April 2017
Editorial Decision:	04 May 2017
Revision received:	04 May 2017
Accepted:	05 May 2017

Transaction Report:

1st Editorial Decision

17 November 2016

Thank you for the submission of your research manuscript to our journal and for your patience while it was seen by three referees. We have now received the full set of referee reports that is copied below.

As you will see, the referees acknowledge the potential interest of the findings. However, all referees also point out that the current dataset does not fully support the conclusions drawn and have a number of suggestions for how the study should be strengthened, and I think that all of them should be addressed. All referees ask for higher quality images (and quantification) showing co-localization and subcellular localization of CRN, CLV2 and BAM3 and to substantiate the conclusion that SERKs are not involved in CLE45 sensing. Moreover, referees 1 and 2 point out missing controls for the BiFC experiments and referee 3 suggests to add a phenotypic analysis of the newly identified bam3 mutants and their response to CLE45.

From the analysis of these comments it becomes clear that significant revision is required before the manuscript becomes suitable for publication in EMBO reports. On the other hand, given the potential interest of your findings and the constructive comments, I would like to give you the opportunity to address the concerns and would be willing to consider a revised manuscript with the understanding that the referee concerns must be fully addressed and their suggestions (as detailed above and in their reports) taken on board.

Should you decide to embark on such a revision, acceptance of the manuscript will depend on a

positive outcome of a second round of review and I should also remind you that it is EMBO reports policy to allow a single round of revision only and that, therefore, acceptance or rejection of the manuscript will depend on the completeness of your responses included in the next, final version of the manuscript.

REFEREE REPORTS

Referee #1:

The manuscript "Perception of root-active CLE peptides requires CORYNE activity in the phloem vasculature" by Hazak et al. reports novel molecular mechanisms in the perception of CLE45 peptide in the Arabidopsis root. The same group has previously shown the genetic role for BAM3 in the perception of CLE45 in the phloem tissue of Arabidopsis root (Depuydt et al, 2013). Based on the report, the authors hypothesized CLE45 and BAM3 act as a ligand-receptor pair. They found novel point-mutation alleles for BAM3 that show insensitivity to CLE45 peptide treatment. One of them, Ser303 was located to the peptide-binding motif within the LRR domain based on the crystal structures for related peptide-receptor complexes. More importantly, the authors have demonstrated the direct interaction of CLE45 and BAM3 LRR domain by using isothermal titration calorimetry (ITC) assays. Analysis on a mutant BAM3 (BAM3QYY) supports the binding of CLE45 peptide to BAM3 LRR domain in a similar fashion to the other peptide-receptor complexes. Since the CLE45 - BAM3 affinity was quite low compared to previously reported CLV3-CLV1 and IDA-HAESA pairs, the authors hypothesized the involvement of co-receptor for the high affinity sensing. They examined candidate receptors, SERK1 and CLV2/CRN complex, by genetic and biochemical assays. Although neither of them seem to work as a co-receptor, the authors found the CRN gene promotes the expression of BAM3 protein at the cell membrane in the developing phloem, and thus it contributes to the perception of CLE45 peptide by BAM3. Even though these findings are interesting in the field of plant peptide signaling, many experiments in the manuscript do not seem to have sufficient quality to support their claims. The details are the following.

Major comments:

1. For all figures, please indicate the sample size (n), especially when s.e.m is used to present the data.
2. Please think about the figure configurations. An important data is presented in supplementary figure (Figure EV2B) while many data in main figure are more likely supplemental (many panels in Figure 2, 3 and 4). The titles for Figure EV1 and Figure EV2 are same.
3. Page 7 "In Arabidopsis, bam3QYY-CITRINE was specifically expressed in the protophloem, with a similar profile of subcellular (plasma membrane) localization and abundance as wild type BAM3-CITRINE (Fig 1D, EV1C)."
The fluorescent signal in BAM3:bam3QYY-CITRINE in Col-0 seems much weaker compared to the wild type BAM3-CITRINE (Figure 1D). Please provide a better image.
4. Page 8 "To test their involvement genetically, we surveyed the response of serk mutants to 14 root-active CLE peptides, including CLE45, which were selected for their significant, reproducible impact on root growth at 50 nM concentration."
-I am concerned about the accuracy of all the peptide treatment assays (Figure 2A to 2D). Please indicate the meaning of the asterisks in column "Col-0" in Figure 2B, "****a" and "****b" in Figure 2C, describe all the peptide sequences used in this study and re-confirm if the results are correct and reproducible. The details are the following.

Page 8 "Indeed, representative serk loss-of-function mutants (alleles serk1-1, serk2-1, serk3-1, serk4-1 and serk5-1) displayed substantial resistance to application of individual CLE peptides, which was partly specific and partly overlapping.
-Which peptides were insensitive or hypersensitive in which mutants? I found many phenotypes are not properly evaluated in Figure 2B, e.g. CLE22 treatment does not affect the root length in WT but serk mutants are indicated to be "resistant" to CLE22.

Page 8 "In the case of CLE6, all five mutants were strongly insensitive (Fig 2A)."
-The activity of CLE6 peptide at 50nM (Figure 2) seems inconsistent with previous reports

including a report from the same group (Kinoshita et al, 2007 Plant Cell Physiol 48:1821; Whitford et al, 2008 PNAS 105:18625; Depuydt et al. 2013 PNAS 110:7074) where CLE6 peptides exert the root shortening activity only at a very high concentration (10 μ M). Results on CLE8, CLE16, CLE17, CLE20 and CLE22 are also different from the previous report (Depuydt et al. 2013).

Page 8 "Finally, in the case of five CLE peptides, all serk mutants behaved like wild type, and in a few cases, individual serk mutants displayed hypersensitivity (Fig 2A-B)."

For which peptides did you see the hypersensitivity? In Figure 2B, serk mutants are described as hypersensitive against CLE17 even though CLE17 treatment does not affect root growth in wild type (Figure 2A). Some results are not consistent between Figure 2A and Figure 2D, e.g. CLE17 and CLE20 show almost similar activities in Figure 2D but CLE17 does not show the activity in Figure 2A.

From the results, differences less than 1cm seem meaningless even if the values are statistically significant. Please carefully check all the data and correct all the statement related to Figure 2A-2D.

5. Page9 "A genetic role of SERK genes in CLE45 perception might be masked by redundancy, and therefore the notion that SERK1 could be a BAM3 co-receptor still appeared attractive..."

-Since only SERK1 was tested for its involvement in CLE45 perception (Figure EV2 and Figure 3), other SERKs may act as a co-receptor for BAM3 in CLE45 perception. The biochemical assay and expression analysis on other SERKs will be needed to clarify this point.

6. Page 10-11 "...we assessed the relative contribution of other known CLV/BAM signaling components, previously analyzed in the shoot meristem (Guo et al, 2010; Meng & Feldman, 2010; Muller et al, 2008; Pallakies & Simon, 2014), CLV2 and CRN (Page 10)... Taken together, our experiments suggest that CLV2 and CRN are necessary to mediate sensitivity to all root-active CLE peptides monitored in this study."

-The role for CRN/SOL2 in root CLE perception has been reported in Muller et al 2008 and Miwa et al Plant Cell Physiol 49:1752.

7. Page 12 "when expressed alone, CLV2 and CRN fusion proteins were found in the endoplasmic reticulum (Fig 4A-B)."

-The localization patterns for CRN and CLV2 look very different (Figure 4A-B). The CRN-TurboRFP seems to be rather localized to plasma membrane with intracellular vesicles (Figure 4A). Please explain. If possible, please use ER marker to clarify this point.

8. Page 12 "While in wild type or the crn background CRN-CITRINE localized to the plasma membrane, it did not accumulate at the plasma membrane in clv2 mutants (Fig 4E-F) (Fig EV3C)."

-The resolution of images is poor. I still see some membrane-localized citrine signals in the clv2 mutant background although it is weaker compared to the intracellular vesicles (Figure 2F'). Membrane localization in the wild type background is not very clear (Figure 2E and 2E').

9. Page 12 "Conversely, CLV2-CITRINE fusion protein expressed under control of the CLV2 promoter displayed plasma membrane localization when expressed in the clv2 mutant background, but diffusive cytoplasmic localization when expressed in the crn mutant (Fig EV3D)."

-Although I find difference between the clv2 and crn mutant backgrounds, it is almost impossible to specify the subcellular localization of CLV2-CITRINE signals due to the low resolution of the photos (Figure EV3A, D). Please provide magnified images with a higher resolution.

10. Page13 "Also, in transient co-expression in tobacco no co-localization was observed for CRN and BAM3 fusion proteins (Fig 4G)."

-It appears that we can detect some co-localization (yellow colored) in the Figure 4G', which means a portion of CRN-TurboRFP is already localized in plasma membrane without additional CLV2 expression. Please clarify.

11. Page 13 "Moreover, a modest but robust BiFC interaction between BAM3 and CRN could be observed when (non-fluorescent) CLV2-HA was co-expressed as well (Fig 4J). Thus, it appears that in principle, BAM3 is capable of interaction with the CLV2-CRN dimer in a cellular setting."

-BiFC is not a quantitative method and is typically combined with other methods, such as FRET, to prove a natural interaction. In BiFC, "even a weak interaction between the test proteins will reconstitute the YFP. Once reconstituted, the YFP halves do not dissociate... Negative controls are

critical for every test pair and for each test session. If the test proteins are membrane proteins, then the negative control proteins also needed to be membrane proteins." quoted from Tunc-Ozdemir et al. 2016 *Methods Mol Biol.* 1363:155. However, the authors did not perform the negative control experiment with membrane localized protein.

Minor comments:

12. Page 3 "Arabidopsis contains 31 CLE genes, some of which encode redundant peptides, thereby giving rise to 26 distinct CLE peptides (Ito et al, 2006)."

-Arabidopsis genome encode 32 CLE genes. CLE43 (Strabala et al, 2006 *Plant Physiol* 140: 1331-1344) is missing in Ito et al 2006.

13. Page 4 "Consistently, it was recently suggested that SERK1 also plays a role in PXY/TDR-mediated CLE41/44/TDIF signal transduction (Zhang et al, 2016a)."

-The reference may be Zhang et al, 2016b.

14. Page 5 "Finally, CLV1 has been implicated in stem cell homeostasis in the root meristem"

-In the root meristem, BAM1 and RPK2 receptor kinases are also involved in CLE signaling (Shimizu et al, 2015 *New Phytol* 208: 1104-1113). Please refer to them.

15. Are the peptides used in the biochemical assays (ITC and gel filtration) bioactive?

16. The presentation of Figure 1B is poor. Please add the magnified view around the binding surface between the peptide and receptor.

17. Page 9 "we could not detect formation of CLE45-dependent BAM3 and SERK1 ectodomain complexes (Fig EV2B)".

-I see two faint bands in the peak1 lane in Fig EV2B. One of them corresponds to the size of SERK1. Please explain.

18. Page 11-12 "Together with the rescue of CLE peptide resistance through protophloem specific CRN expression, these observations indicate that the developing phloem is indeed a crucial site of action for root-efficient CLE peptides..."

-Using CRN transgene driven under BAM3 promoter, the authors show the importance of phloem for CLE peptide sensing. However, since BAM3 has been reported to sense CLE peptide in the phloem, the results seem to just support their previous finding.

19. Page 13 "Such interaction could actually occur in planta, since both BAM3 and CRN displayed polar localization at the shootward plasma membrane of developing sieve elements (Fig 4K-L)."

-Did you observe similar subcellular localization of BAM3:CRN-CITRINE in WT (Figure 4E)?

20. Page 13 "Similarly, crn second site mutation substantially rescued the protophloem differentiation, root meristem size and root growth defects of brx mutants (Fig 5A-C)."

-Please provide better magnified images for Figure 5C.

21. The order of the legends for Figure 3B/C-D seems wrong (Page 27).

Referee #2:

Hazak et al describe their biochemical and genetic efforts to identify the CLE45 binding site. Genetic data suggested BAM3 to be a receptor for this peptide, but biochemical support was lacking. Here, using ITC, structural homology modeling and site-directed mutagenesis, the authors provide firm support that BAM3 is a binding site for CLE45, and that its CLE45 binding contributes to biological function. Since peptide binding affinity is low, the authors next search for potential co-receptors and provide evidence that the SERK proteins are not involved. Finally, the authors provide support for roles of the CLV2 and CRN proteins in the BAM3-dependent CLE45 response. All in all, the work is convincing and addresses important questions in plant peptide signaling. There are several rough edges and leaps of faith that I would propose the authors to address for this manuscript to be as strong as it could be.

Specific comments (in random order):

1. The Kd of BAM3-CLE45 interaction is surprisingly high, and it is difficult to imagine how high micromolar peptide concentrations could be generated to satisfy binding in the apoplast. The assumption is that this is because a co-receptor is missing, and much of the manuscript focuses on characterizing candidates for such a co-receptor function. After exploring the SERK family (negative), the authors focus on CRN and CLV2, and provide evidence that both are required for CLE45 perception by BAM3. However, the role as co-receptor remains purely speculative, and in fact not well supported by the data (as the authors conclude). What I miss in the paper is a critical reflection on the low affinity, including alternative hypothesis versus the missing co-receptor idea. Is it possible that the affinity is actually this low? Could BAM3 lack post-translation modifications in the ectodomain that modify affinity? Could the conformation of the protein outside the ectodomain modify affinity?
2. The positive evidence for the interaction between BAM3 and CRN/CLV2 is BiFC data in Figure 4. However, this needs to be better documented and quantified. Now, the only evidence is shown in Figure 4J', and the quality of that panel is questionable. The same is true for the proposed co-localization (comparison Figure 4K/L), which is far from convincing.
3. The root images in Figure 1D are not very clear, and it is difficult to spot the BAM3 signal. It could help to use magenta/green colors, or false-color the Citrine signal on a greyscale background.
4. The statement that BAM3 "operates independent of SERK proteins" is based on negative evidence (which is unavoidable). The lack of biochemical evidence for (CLE45-induced) interaction between BAM3 and SERK1 ectodomains and the absence of a clear mutant phenotype (CLE45 resistance) in *serk1*. However, this does not exclude that any other combination of 2 or more SERK proteins could act as co-receptor. To make such a firm statement, the authors should at least show that the ectodomain of one other SERK protein also does not interact with the BAM3 ectodomain in the presence of CLE45.
5. From the images provided in Figure 4A, it is not clear that SERK1-Citrine is not expressed in the protophloem. These panels need better labeling and counterstaining.
6. Reduced BAM3-Citrine accumulation on *crn* roots (Figure 5F/G) is impossible to verify in the absence of quantification of fluorescence signals.

Referee #3:

In the manuscript entitled 'Perception of root-active CLE peptides requires CORYNE activity in the phloem vasculature', evidences supporting the idea that BAM3 is a receptor of CLE45 are provided. Amino acid residues that are crucial for the activity as well as its binding are identified. It is intriguing that CLV2/CRN regulates BAM3 protein expression and thereby regulates phloem development. Besides, this ligand/receptor regulation does not seem to require SERK family proteins as co-receptors. Although the authors presented nice experimental data to support their idea, we have some suggestions that can potentially make the manuscript more coherent and concrete.

Major comments:

1. The fact that SERK1 is not a part of the CLE45 and BAM3 complex might be a strong evidence to support that SERK1 is not a co-receptor. The authors' somewhat conflicting result on *serk1-1* loss of function mutant might be in fact due to an unknown second site mutation. Although it might be highly relevant to map where the second mutation is, it might not dramatically improve the quality of the manuscript. Alternatively, in order to make the manuscript more consistent, we suggest that the authors do the same treatment on the other allele *serk1-3*, not only CLE45 but also other CLEs. *serk1-1* aspect might well suit in supplementary data.
2. Authors stated in the discussion that "CLV2 and CRN are part of all root-active CLE peptide signalling pathways, possibly by controlling the expression, proper membrane localization, and/or stability of LRR-RK signalling complexes". However, there is still a possibility that BAM3 expression is regulated transcriptionally, or it could be by stabilizing BAM3 protein or combination

of both. We suggest that the authors observe transcriptional reporter lines to see the transcript level of BAM3 in *crn* mutant background. Or alternatively to measure mRNA level by qRT-PCR. This will at least resolve if it's transcriptional regulation or not.

3. One way to bolster their statement about CRN's BAM3 regulation would be to do a simple complementation. If authors' already generated CRN::CRN in *crn* mutant background, we highly recommend that the authors use this line as well. BAM3-BAM3-CITRINE construct are introduced into the *crn* mutant and are compared with the transgenic line with the same construct introduced into Col-0. This indicates the insertion site of the construct is different. It is hard to compare the fluorescent signal between different insertion lines.

4. It is interesting that the expression of CRN is enriched in the SE, although the expression is not exclusive. However, current confocal images don't seem to well support or emphasize this aspect. Therefore, we suggest that the authors provide high mag. cross section images and also quantitative data if possible. Along these lines, we'd like to ask the authors to do the followings.

4-1. to replace the image of pMAKR5::MAKR5-GFP in figure 3 with the CRN reporter line. The text referred to MAKR5 only later, but not in the context of figure 3.

4-2. to provide higher magnification images in Fig 3, as the sub-cellular localization of CRN and CLV2 under their native promoters cannot be distinguished in the provided images.

4-3. In Fig 1D, *bam3*QYY-CIRTRIN in Col-0 looks weaker than BAM3-CITRINE. Could you replace the images more typical and easy to compare?

5. From their genetic screening, they isolated new *bam3* alleles. However, they only described the mutation site in the BAM3 gene and did not show any phenotypic analysis. It would be good to show if the CLE45 response of these mutants are the same or not (is some of them are weak allele?).

6. We'd also like the authors to answer or discuss following aspects in the discussion part.

6-1. How do you explain the insensitivity of *crn* to all CLE peptides? Does it control expression of CLV as well?

6-2. CLV3 peptide they used are not glycopeptide. It is better to use glyco-modified CLV3 peptide to state that BAM3 is not a receptor of CLV3. It is possible the BAM3 has a binding activity of the other CLEs because BAM3 has been shown to have a binding activity to CLE9 in a previous report (Shinohara et al. Plant J. 2012).

6-3. Please elaborate why *crn* only partially rescues *brx* mutant, whereas *bam3* fully rescues it, if *crn* is absolutely required for BAM3 function.

6-4. The authors state that *crn* mutants do not display an apparent morphological root phenotype, which suggests that none of the CLEs are required for normal root development. Could their purpose be investigated a bit further? Are they induced by some environmental factors then?

Minor comments:

p.19

More detailed description of isolation of *crn-10* allele. TILLING? Mutant screening?

p.3

Would you write every "root-active CLEs" in the main text to make easy to read? (For example, root-active CLEs; CLE6, 8, 9/10, and 45)

p.6

The sentence ' Root protophloem differentiation and thereby root growth of *bam3* loss-of-function mutants is not impaired by exogenously applied CLE45, suggesting that BAM3 could act as a CLE45 receptor (Depuydt et al, 2013). ' is a strong expression. The *bam3* mutants reduced their primary root length by CLE45 but the reduction is quite milder than wild-type. This suggests there are BAM3-independent CLE45 perception pathways, which would explain why the *bam3* mutant does not have any obvious phenotype in root development.

-we suggest that the authors change the labelling of Figure 1F? It could easily be simplified (e.g. using only one letter repeatedly instead of 2 numbers).

We now submit the revised version of our manuscript "*Perception of root-active CLE peptides requires CORYNE activity in the phloem vasculature.*" In this substantial revision, we have addressed the reviewers' comments and added various new data, which are displayed in the (revised) Figures 1B-C, 2B-C, 5A-B, 5G-J, EV1A-C, EV1F, EV2F-G, EV3C, EV4A-I, and EV5.

The key revisions are as follows:

1. We determined that indeed as suspected by one reviewer, N-terminally modified CLE peptides largely lose their biological activity. Because this put our original biochemical data in a new perspective, we repeated all binding assays. The new data, using native peptide sequences, now reveal substantially stronger BAM3-CLE45 binding, in the nanomolar affinity range, corroborating that BAM3 is indeed a *bona fide* CLE45 receptor.
2. We also repeated the ITCs and gel filtrations with SERK1, but again found no interaction between BAM3 and SERK1, whether in the presence or absence of CLE45. We have performed similar experiments with SERK3 now, with the same result.
3. We have performed biological assays of the *serk1-3* allele against all CLE peptides monitored in our study, and found no resistance to any. Moreover, we have analyzed the *serk1-1* allele by whole genome sequencing and found that it carries a *bam3* null mutation, which we show is responsible for the CLE45 resistance of *serk1-1*.

In summary, point 1 considerably strengthens our original proposal that BAM3 is the CLE45 receptor, while points 2 and 3 confirm our previous conclusion that SERKs likely do not play a role in CLE45 perception. Other revisions include improved data presentations and additional experiments, all of which confirm our initial observations and strengthen our conclusions. Please find a point-by-point response to the reviewers below.

POINT-BY-POINT RESPONSE

Referee #1:

Major comments:

"1. For all figures, please indicate the sample size (n), especially when s.e.m is used to present the data."

We apologize for this omission; sample sizes are now indicated wherever applicable.

"2. Please think about the figure configurations. An important data is presented in supplementary figure (Figure EV2B) while many data in main figure are more likely supplemental (many panels in Figure 2, 3 and 4). The titles for Figure EV1 and Figure EV2 are same."

Figures and panels have now been re-arranged to better highlight the key points. Figure legends have been carefully reviewed for accuracy. A new version of Fig EV2B using a CLE45 peptide lacking an N-terminal tyrosine residue (see below) is now shown in Fig 2B, as requested by the reviewer.

"3. Page 7 "In *Arabidopsis*, *bam3QYY-CITRINE* was specifically expressed in the protophloem, with a similar profile of subcellular (plasma membrane) localization and abundance as wild type *BAM3-CITRINE* (Fig 1D, EV1C)." The fluorescent signal in *BAM3:bam3QYY-CITRINE* in Col-0 seems much weaker compared to the wild type *BAM3-CITRINE* (Figure 1D). Please provide a better image."

We now provide better images (new Fig 1D) as requested.

"4. Page 8 "To test their involvement genetically, we surveyed the response of *serk* mutants to 14 root-active CLE peptides, including CLE45, which were selected for their significant, reproducible

*impact on root growth at 50 nM concentration." -I am concerned about the accuracy of all the peptide treatment assays (Figure 2A to 2D). Please indicate the meaning of the asterisks in column "Col-0" in Figure 2B, "***a" and "***b" in Figure 2C, describe all the peptide sequences used in this study and re-confirm if the results are correct and reproducible. The details are the following." ...and subordinate comments.*

We thank the reviewer for taking a careful look at these data.

- There was indeed a transmission/labeling error, CLE6 was not assayed (not root-active, as outlined by this reviewer), but CLE25 was, and we have now carefully reviewed the raw data and reassembled the figure (Fig 2A). We have also redone the assays for all peptides with the *serk1-3* allele and replaced the *serk1-1* data. To facilitate direct comparison, we combined the data for all *serk* mutants and *clv2* as well as *crn* in Fig 2A. Finally, we sequenced CLE45-resistant plants segregating from a *serk1-1* cross by whole genome sequencing and found that the *serk1-1* allele carries a *bam3* null mutation that is responsible for its CLE45 resistance (line 179).

- The 14 root-active CLE peptides are now spelled out in the text (line 170), and their sequences, including those of two non-root-active control peptides, are given in the Methods.

- Differences to published data partly reflect variation between replicate experiments, but in most cases (i.e. as compared to Depuydt et al. 2013) the fact that 50 nM concentration was used, instead of 100 nM.

- We agree with the reviewer that effect sizes were typically small, and in this sense the table we had provided was misleading. We have now deleted this figure panel, and changed our wording to properly reflect that none of the individual *serk* mutants displayed marked resistance to any of the CLE peptides (line 172), which is also in line with the new data on the *serk1-3* mutant.

"5. Page9 "A genetic role of SERK genes in CLE45 perception might be masked by redundancy, and therefore the notion that SERK1 could be a BAM3 co-receptor still appeared attractive..." -Since only SERK1 was tested for its involvement in CLE45 perception (Figure EV2 and Figure 3), other SERKs may act as a co-receptor for BAM3 in CLE45 perception. The biochemical assay and expression analysis on other SERKs will be needed to clarify this point."

At this point, we are unfortunately not in a position to deliver high-resolution expression or biochemical analyses for all *SERKs*, which in our opinion would also go beyond the scope of this manuscript. However, in the revised manuscript, we now present new biochemical data that suggest that both *SERK1* and *SERK3* (BAK1) cannot act as co-receptor kinases for *BAM3*-CLE45. Analytical size-exclusion chromatography assays with the *SERK3* ectodomain are now presented in Fig EV2F. We have revised our manuscript, which now states (line 193): "However, although the *BAM3* and *SERK1* kinase domains were able to trans-phosphorylate each other in an *in vitro* kinase assay (Fig EV2E), neither *SERK1* nor *SERK3* formed CLE45-dependent complexes with *BAM3* *in vitro* (Fig 2B, EV2F)." We could also not define a genetic role for *SERK3* in the sensing of root active CLE peptides (Fig 2A).

"6. Page 10-11 "...we assessed the relative contribution of other known CLV/BAM signaling components, previously analyzed in the shoot meristem (Guo et al, 2010; Meng & Feldman, 2010; Muller et al, 2008; Pallakies & Simon, 2014), CLV2 and CRN (Page 10)... Taken together, our experiments suggest that CLV2 and CRN are necessary to mediate sensitivity to all root-active CLE peptides monitored in this study." -The role for CRN/SOL2 in root CLE perception has been reported in Muller et al 2008 and Miwa et al Plant Cell Physiol 49:1752."

Indeed, thank you for pointing this out, we apologize for omitting these references, which have now been added.

"7. Page 12 "when expressed alone, CLV2 and CRN fusion proteins were found in the endoplasmic reticulum (Fig 4AB)." -The localization patterns for CRN and CLV2 look very different (Figure 4A-B). The CRN-TurboRFP seems to be rather localized to plasma membrane with intracellular vesicles (Figure 4A). Please explain. If possible, please use ER marker to clarify this point."

The reviewer is right that the ER-localization of CRN-TurboRFP is not very clear in the picture, mostly because in our experience, Turbo-RFP fusion always leads to some ER aggregation. We have however repeated the experiments with an alternative mTFP1 tag and also included ER-markers. All of these complementary and confirmatory data are now displayed in the new Figure EV4.

“8. Page 12 "While in wild type or the crn background CRN-CITRINE localized to the plasma membrane, it did not accumulate at the plasma membrane in clv2 mutants (Fig 4E-F) (Fig EV3C)." -The resolution of images is poor. I still see some membrane-localized citrine signals in the clv2 mutant background although it is weaker compared to the intracellular vesicles (Figure 2F'). Membrane localization in the wild type background is not very clear (Figure 2E and 2E').”

We assume that the reviewer refers to panels in Figure 4 here. We have now replaced the images with new ones, which are hopefully clearer. We also reworded the text to indicate that plasma membrane abundance is substantially reduced, but not completely abolished (lines 266, 269).

“9. Page 12 "Conversely, CLV2-CITRINE fusion protein expressed under control of the CLV2 promoter displayed plasma membrane localization when expressed in the clv2 mutant background, but diffusive cytoplasmic localization when expressed in the crn mutant (Fig EV3D)." -Although I find difference between the clv2 and crn mutant backgrounds, it is almost impossible to specify the subcellular localization of CLV2-CITRINE signals due to the low resolution of the photos (Figure EV3A, D). Please provide magnified images with a higher resolution.”

Here we also provide new, better pictures as well as magnifications that are hopefully clearer now.

“10. Page13 "Also, in transient co-expression in tobacco no co-localization was observed for CRN and BAM3 fusion proteins (Fig 4G)." -It appears that we can detect some co-localization (yellow colored) in the Figure 4G', which means a portion of CRN-TurboRFP is already localized in plasma membrane without additional CLV2 expression. Please clarify.”

We assume the reviewer refers to Fig 4G'. We do not see any clear yellow fluorescence there, but maybe it is possible that a small fraction of CRN makes it to the plasma membrane because of internal CLV2. In any case, as shown in Figure 4H, co-expression of CLV2 has a massive effect on the co-localization.

“11. Page 13 "Moreover, a modest but robust BiFC interaction between BAM3 and CRN could be observed when (nonfluorescent) CLV2-HA was co-expressed as well (Fig 4J). Thus, it appears that in principle, BAM3 is capable of interaction with the CLV2-CRN dimer in a cellular setting." -BiFC is not a quantitative method and is typically combined with other methods, such as FRET, to prove a natural interaction. In BiFC, "even a weak interaction between the test proteins will reconstitute the YFP. Once reconstituted, the YFP halves do not dissociate... Negative controls are critical for every test pair and for each test session. If the test proteins are membrane proteins, then the negative control proteins also needed to be membrane proteins." quoted from Tunc-Ozdemir et al. 2016 Methods Mol Biol. 1363:155. However, the authors did not perform the negative control experiment with membrane localized protein.”

We agree with the reviewer that BiFC is tricky and parallel control experiments have to be performed. This is what we had done, but we decided to not include all controls in order to not overload the figure. This was obviously a mistake however, and we now provide MAKR5 as negative control for BAM3, as well as BRI1 as a negative control for CLV2-CRN, and a positive control for BRI1, BKI1 (line 284; Fig 4K-M).

Minor comments:

“12. Page 3 "Arabidopsis contains 31 CLE genes, some of which encode redundant peptides, thereby giving rise to 26 distinct CLE peptides (Ito et al, 2006)." -Arabidopsis genome encode 32 CLE genes. CLE43 (Strabala et al, 2006 Plant Physiol 140: 1331-1344) is missing in Ito et al 2006.”

Thank you, this has been corrected and references have been adjusted (line 49).

"13. Page 4 "Consistently, it was recently suggested that SERK1 also plays a role in PXY/TDR-mediated CLE41/44/TDIF signal transduction (Zhang et al, 2016a)." -The reference may be Zhang et al, 2016b."

Thank you; this has been corrected.

"14. Page 5 "Finally, CLV1 has been implicated in stem cell homeostasis in the root meristem" -In the root meristem, BAM1 and RPK2 receptor kinases are also involved in CLE signaling (Shimizu et al, 2015 New Phytol 208: 1104-1113). Please refer to them."

Yes, indeed, we have now amended our introduction with this reference (line 99).

"15. Are the peptides used in the biochemical assays (ITC and gel filtration) bioactive?"

We are grateful to reviewer #1 for suggesting this experiment: We had added an N-terminal tyrosine residue to our CLE45, CLE41/44 and CLV3 peptides, in order to accurately determine the peptide concentrations in solution via UV absorbance. While addition of an N-terminal tyrosine to the sequence-related IDA/IDL peptides did not affect their bioactivity or alter their binding properties to the receptor kinase HAESA (Santiago et al., eLife, 2016), the root growth assays now presented in Fig EV1F clearly show that addition of an N-terminal Tyr renders different CLE peptides inactive. We thus repeated our entire biochemical analyses using non-modified CLE peptides, the results of which are discussed below in the response to reviewer #2 (point 1). Again, thank you for pointing this issue out to us. A statement has been included in the manuscript, which reads (lines 137-140) "Importantly, addition N-terminal extension of CLE45 or CLV3 by a tyrosine residue (initially used to quantify the peptide concentrations) rendered the engineered peptides non-bioactive and drastically reduced binding to the BAM3 ectodomain (Fig EV1D-F)." We describe an alternative assay to measure CLE peptide concentrations in the revised method section, accordingly.

"16. The presentation of Figure 1B is poor. Please add the magnified view around the binding surface between the peptide and receptor."

We now provide the desired magnified view of the modeled peptide-binding surface in BAM3 (Fig 1B).

"17. Page9 "we could not detect formation of CLE45-dependent BAM3 and SERK1 ectodomain complexes (Fig EV2B)". -I see two faint bands in the peak1 lane in Fig EV2B. One of them corresponds to the size of SERK1. Please explain."

The weak band corresponds to a BAM3 degradation product as identified by in gel digest followed by mass spectrometry. We have repeated the experiments with freshly prepared BAM3. The results are shown in revised Fig 2B.

"18. Page 11-12 "Together with the rescue of CLE peptide resistance through protophloem specific CRN expression, these observations indicate that the developing phloem is indeed a crucial site of action for root-efficient CLE peptides..." -Using CRN transgene driven under BAM3 promoter, the authors show the importance of phloem for CLE peptide sensing. However, since BAM3 has been reported to sense CLE peptide in the phloem, the results seem to just support their previous finding."

This must be a misunderstanding. It is true that we have shown that BAM3 is required for CLE45 perception in the protophloem, but here we show that CRN activity in the protophloem is not only required for CLE45 perception, but also for perception of all other root-active CLE peptides tested here.

"19. Page 13 "Such interaction could actually occur in planta, since both BAM3 and CRN displayed polar localization at the shootward plasma membrane of developing sieve elements (Fig 4K-L)." - Did you observe similar subcellular localization of BAM3:CRN-CITRINE in WT (Figure 4E)?"

Yes, the localization is the same in wild type. We have now amended the figure (4N) to show this.

“20. Page 13 “Similarly, *crn* second site mutation substantially rescued the protophloem differentiation, root meristem size and root growth defects of *brx* mutants (Fig 5A-C).” -Please provide better magnified images for Figure 5C.”

We have replaced the images with a new set that is hopefully clearer.

“21. The order of the legends for Figure 3B/C-D seems wrong (Page 27).”

Thank you, this has been fixed.

Referee #2:

Specific comments (in random order):

“1. The *K_d* of BAM3-CLE45 interaction is surprisingly high, and it is difficult to imagine how high micromolar peptide concentrations could be generated to satisfy binding in the apoplast. The assumption is that this is because a coreceptor is missing, and much of the manuscript focuses on characterizing candidates for such a co-receptor function. After exploring the SERK family (negative), the authors focus on CRN and CLV2, and provide evidence that both are required for CLE45 perception by BAM#. However, the role as co-receptor remains purely speculative, and in fact not well supported by the data (as the authors conclude). What I miss in the paper is a critical reflection on the low affinity, including alternative hypothesis versus the missing co-receptor idea. Is it possible that the affinity is actually this low? Could BAM3 lack post-translation modifications in the ectodomain that modify affinity? Could the conformation of the protein outside the ectodomain modify affinity?”

Based on a suggestion made by reviewer #1 (see above, point 15), we performed bioassays with the CLE peptides used for the biochemical experiments. We found that addition of an N-terminal Tyr residue renders CLE45 and CLV3 inactive in planta (new Fig EV1F). We have thus repeated all biochemical experiments presented in the manuscript using non-modified peptides, which are active in our root growth bioassay (new Fig EV1F, revised Fig 2A). We found that while Y-CLE45 binds the BAM3 ectodomain with micromolar affinity, native CLE45 is sensed with a *K_d* of approx. 100 nM (revised Fig 1C). Thus, we can now present high affinity binding of CLE45 to BAM3. Binding is specific, as BAM3 binds the non-root CLE peptide CLV3 with a *K_d* of approx. 10 micromolar (new Fig EV1C). Binding of CLE45 to BAM3 is comparable to the high-affinity binding of CLE41/44 to the CLE receptor PXY (Fig EV1B; approx. 10 nM in our hands, 30 nM in Zhang et al., 2016a). We have updated Fig 1C, 2C, Fig EV1 B-E, EV2 F, G and revised the manuscript accordingly. It now reads (lines 132) “We found that BAM3 bound CLE45 with a *K_d* of ~120 nM and with 1:1 stoichiometry (Fig 1C). The binding affinity for CLE45 to BAM3 was about 10-fold lower than CLE41/44 binding to the LRR ectodomain of PXY (Fig EV1B) (Zhang *et al.*, 2016a). BAM3 showed specific CLE45 binding, as the sequence-related CLV3 peptide, which is not expressed in the root (Fiers *et al.*, 2005), bound with much lower affinity (*K_d* ~10 μM) (Fig EV1C). Importantly, addition N-terminal extension of CLE45 or CLV3 by a tryrosine residue (initially used to quantify the peptide concentrations) rendered the engineered peptides non-bioactive and drastically reduced binding to the BAM3 ectodomain (Fig EV1D-F).” Importantly, the mapping of the BAM3 CLE binding surface remains valid (lines 143): “In our homology model, BAM3 residues Q226, Y228, and Y231 from the LRR domain form part of the CLE45 binding surface. Consistently, binding of CLE45 to a purified BAM3 ectodomain in which Q226, Y228 and Y231 were mutated to alanines (BAM3QYY) was ~8 times weaker when compared to the wild type LRR domain (Fig 1C).”

Also, despite the high-affinity CLE45 binding to BAM3, we cannot find a CLE45-dependent interaction of BAM3 with SERK1 (or SERK3, see below) (revised Fig 2B-C, EB2F). This is in sharp contrast to the CLE receptor PXY and its ligand CLE41/44, which forms complexes with SERK1 (our data are shown in Fig EV2G, compare Zhang et al, 2016b)

“2. The positive evidence for the interaction between BAM3 and CRN/CLV2 is BiFC data in Figure 4. However, this needs to be better documented and quantified. Now, the only evidence is shown in Figure 4J, and the quality of that panel is questionable. The same is true for the proposed co-localization (comparison Figure 4K/L), which is far from convincing.”

Please refer to our comments to the editor and reviewer 1 above. We assume the reviewer refers to panel Fig 4J". This BiFC was robust, and we have now added additional controls (Fig 4K-M) as well as co-localization assays (new Fig EV4) as outlined above.

"3. The root images in Figure 1D are not very clear, and it is difficult to spot the BAM3 signal. It could help to use magenta/green colors, or false-color the Citrine signal on a greyscale background."

Indeed, the images were suboptimal; we have now replaced them and used magenta background for better visualization in a number of panels, as suggested.

"4. The statement that BAM3 "operates independent of SERK proteins" is based on negative evidence (which is unavoidable). The lack of biochemical evidence for (CLE45-induced) interaction between BAM3 and SERK1 ectodomains and the absence of a clear mutant phenotype (CLE45 resistance) in serk1. However, this does not exclude that any other combination of 2 or more SERK proteins could act as co-receptor. To make such a firm statement, the authors should at least show that the ectodomain of one other SERK protein also does not interact with the BAM3 ectodomain in the presence of CLE45."

As requested we now present interaction studies using SERK1 and SERK3 (BAK1) as co-receptor candidates. In agreement with our finding that neither SERK1 nor SERK3 mutants display strong CLE-resistant phenotypes in our root assay (Fig. 2A), we cannot detect CLE45-dependent complex formation between BAM3 and either SERK1 (Fig.- 2B-C) or SERK3 (Fig.- EV2F). The revised manuscript reads (lines 193): "However, although the BAM3 and SERK1 kinase domains were able to trans-phosphorylate each other in an *in vitro* kinase assay (Fig EV2E), neither SERK1 nor SERK3 formed CLE45-dependent complexes with BAM3 *in vitro* (Fig 2B, EV2F)."

"5. From the images provided in Figure 4A, it is not clear that SERK1-Citrine is not expressed in the protophloem. These panels need better labeling and counterstaining."

In light of the identified *bam3* loss-of-function background mutation, the *SERK1* expression pattern is not as pertinent as before, but we have provided better, hopefully clearer images (Figure EV2H).

"6. Reduced BAM3-Citrine accumulation on crn roots (Figure 5F/G) is impossible to verify in the absence of quantification of fluorescence signals."

The difference between wild type and *crn* mutant background is relatively strong and robust. We have observed it across multiple independent transgenic lines (as all observations reported for transgenic constructs in our manuscript). To quantify the fluorescence signal along the protophloem in this contexts is difficult however, because the difference is also position-dependent. That is, whereas closer to the stem cells BAM3 signal is still relatively strong in *crn*, it fades rapidly towards the protophloem transition zone. We now exploited this fact to offer an alternative quantification, by indicating the last cell relative to the first transition zone cell in which BAM3 signal was still detectable (Fig 5J). Whereas in wild type in practically all cases the last cell still clearly expressed BAM3, its expression had already ceased in *crn* 5-6 cell earlier. We hope that this type of quantification illustrates the point. Moreover, to show the effect for individual transgenes, we now also performed crosses between individual lines expressing BAM3-CITRINE in *crn* background to a *crn* line complemented with a CRN transgene. In the F1, BAM3 expression was restored to the wild type level and pattern, corroborating the effect (Fig 5H-I).

Referee #3:

Major comments:

"1. The fact that SERK1 is not a part of the CEL45 and BAM3 complex might be a strong evidence to support that SERK1 is not a co-receptor. The authors' somewhat conflicting result on serk1-1 loss of function mutant might be in fact due to an unknown second site mutation. Although it might be highly relevant to map where the second mutation is, it might not dramatically improve the quality of the manuscript. Alternatively, in order to make the manuscript more consistent, we suggest that

the authors do the same treatment on the other allele serk1-3, not only CLE45 but also other CLEs. serk1-1 aspect might well suit in supplementary data."

Please refer to our comments to the editor and reviewers above. We have repeated all assays with *serk1-3* and found no resistance. We have also analyzed *serk1-1* by whole genome sequencing to find that a *bam3* knock out in the background is responsible for its CLE45 resistance (line 179; Fig EV2C-D).

"2. Authors stated in the discussion that "CLV2 and CRN are part of all root-active CLE peptide signalling pathways, possibly by controlling the expression, proper membrane localization, and/or stability of LRR-RK signalling complexes".

However, there is still a possibility that BAM3 expression is regulated transcriptionally, or it could be by stabilizing BAM3 protein or combination of both. We suggest that the authors observe transcriptional reporter lines to see the transcript level of BAM3 in crn mutant background. Or alternatively to measure mRNA level by qRT-PCR. This will at least resolve if it's transcriptional regulation or not."

Indeed, we have now measured *BAM3* expression in wild type and *crn* and found no difference (Fig 5G).

"3. One way to bolster their statement about CRN's BAM3 regulation would be to do a simple complementation. If authors' already generated CRN::CRN in crn mutant background, we highly recommend that the authors use this line as well. BAM3-BAM3-CITRINE construct are introduced into the crn mutant and are compared with the transgenic line with the same construct introduced into Col-0. This indicates the insertion site of the construct is different. It is hard to compare the fluorescent signal between different insertion lines."

Please refer to our comments to reviewer 2 above. Of course, all conclusions are supported by observations in multiple independent transgenic lines for each genotype. Moreover, to show the effect for individual transgenes, we now performed crosses between lines expressing *BAM3-CITRINE* in *crn* background to a *crn* line complemented with a *CRN* transgene. In the F1, *BAM3* expression was restored to the wild type level and pattern (Fig 5H-I).

"4. It is interesting that the expression of CRN is enriched in the SE, although the expression is not exclusive. However, current confocal images don't seem to well support or emphasize this aspect. Therefore, we suggest that the authors provide high mag. cross-section images and also quantitative data if possible. Along these lines, we'd like to ask the authors to do the followings."

"4-1. to replace the image of pMAKR5::MAKR5-GFP in figure 3 with the CRN reporter line. The text referred to MAKR5 only later, but not in the context of figure 3."

We have now replaced the image with an image of the *CRN* reporter line.

"4-2. to provide higher magnification images in Fig 3, as the sub-cellular localization of CRN and CLV2 under their native promoters cannot be distinguished in the provided images."

We now provide better images in Figure EV3, including magnifications that hopefully illustrate this point better now.

"4-3. In Fig 1D, bam3^{QYY}-CIRTRIN in Col-0 looks weaker than BAM3-CITRINE. Could you replace the images more typical and easy to compare?"

We now have replaced those images with better ones.

"5. From their genetic screening, they isolated new bam3 alleles. However, they only described the mutation site in the BAM3 gene and did not show any phenotypic analysis. It would be good to show if the CLE45 response of these mutants are the same or not (is some of them are weak allele?)."

We assayed all of those alleles on different CLE45 concentrations and found them all to be resistant. Those new data are displayed in Figure EV1A.

“6. We'd also like the authors to answer or discuss following aspects in the discussion part.”

“6-1. How do you explain the insensitivity of *crn* to all CLE peptides? Does it control expression of CLV as well?”

At this point we can only speculate on this. As we highlight in the discussion, maybe the expression of other, yet to be identified CLE receptors in the root also depends on *CRN*.

“6-2. CLV3 peptide they used are not glycopeptide. It is better to use glyco-modified CLV3 peptide to state that BAM3 is not a receptor of CLV3. It is possible the BAM3 has a binding activity of the other CLEs because BAM3 has been shown to have a binding activity to CLE9 in a previous report (Shinohara *et al.* Plant J. 2012).”

We thank the reviewer for pointing out the fact that the CLV3 peptide used in the in vitro experiments and the root growth assays is not glycol-modified. The glycosylation of CLV3 has an effect for the affinity the binding to the receptor. For BAM3 however, we used the non-glycosylated CLV3 peptide as a negative control for our ITC experiments. There are several reasons why we think that this control is appropriate which are listed below.

1. The effect of non-glycosylated CLV3 on the root length is similar in WT plants when compared to *bam3* mutant plants which indicates that for this phenotype BAM3 is not sensing the CLV3 peptide (Depuydt *et al.*, 2013).
2. Also, the role of arabinosylation of peptide ligands on the binding affinity to receptors is not clear yet: For CLE9 binding to BAM1 the affinities of arabinosylated and non-arabinosylated CLE9 are similar (CLE9gp and CLE9p interact with BAM1-with K_d of 2.2 and 1.8 nM, respectively; Shinohara *et al.*, 2012). In the same microsomal fractions based binding assay the nonarabinosylated 12-mer version of CLV3 (the one we are using) binds to CLV1 with an affinity of 24 nM while the mature arabinosylated 13-mer CLV3 bind to CLV1 with an affinity of 1 nM (Ohyama *et al.*, 2009). Up to date there are no reports about the structural basis of glycosylated CLE peptide binding to their cognate receptors which makes it difficult to understand the molecular basis of this peptide modification.
3. The binding of CLE9 to BAM3 (Shinohara *et al.*, 2012) mentioned by the reviewer could play a role in other signal transductions pathways. Similar to the situation for CLV3, that CLE9 induced root shortening is not suppressed in *bam3* mutant plants (Depuydt *et al.*, 2013) indicates that CLE9 binding to BAM3 does not play a role in this particular phenotype. We would have liked to extend our BAM3 binding studies to modified versions of CLV3, CLE9 and other CLE peptides, but we are facing difficulties to produce sufficient amounts of BAM3 using our insect cell expression system. We require several large scale preparations (each derived from 12 L of insect cell medium) to obtain sufficient material for one ITC experiment plus appropriate controls. This means that one assay takes about 4 weeks to accomplish and costs about 10K EUR. We are thus presently unable to perform quantitative binding assays of BAM3 with many different CLE peptides. In our revised manuscript we now present high affinity, specific binding of BAM3 to CLE45 (Fig 1B), whereas CLV3 is bound only weakly (Fig EV1C, a 70fold reduction in binding affinity). The revised manuscript reads (line 132): “We found that BAM3 bound CLE45 with a K_d of ~120 nM and with 1:1 stoichiometry (Fig 1C). The binding affinity for CLE45 to BAM3 was about 10-fold lower than CLE41/44 binding to the LRR ectodomain of PXY (Fig EV1B) (Zhang *et al.*, 2016a). BAM3 showed specific CLE45 binding, as the sequence-related CLV3 peptide, which is not expressed in the root (Fiers *et al.*, 2005), bound with much lower affinity (K_d ~10 μ M) (Fig EV1C). Importantly, addition Nterminal extension of CLE45 or CLV3 by a tryrosine residue (initially used to quantify the peptide concentrations) rendered the engineered peptides non-bioactive and drastically reduced binding to the BAM3 ectodomain (Fig EV1D-F).”

“6-3. Please elaborate why *crn* only partially rescues *brx* mutant, whereas *bam3* fully rescues it, if *crn* is absolutely required for BAM3 function.”

We believe that this is a misunderstanding because of a matter of semantics. By saying that *crn* is “absolutely” required, we mean that without *CRN*, CLE45 sensitivity is not as in wild type, but we also state that it is not zero as compared to *bam3*. We now elaborate on this in the discussion and have reworded to avoid ambiguity. Along these lines, it is also coherent then that *brx* is only partially rescued.

“6-4. The authors state that *crn* mutants do not display an apparent morphological root phenotype, which suggests that none of the CLEs are required for normal root development. Could their purpose be investigated a bit further? Are they induced by some environmental factors then?”

Unfortunately, we can only speculate at this point, but we have extended our discussion to highlight possible scenarios, including genetic redundancy as *crn* is not fully CLE45 insensitive (see above) (line 406).

Minor comments:

“p.19 More detailed description of isolation of *crn-10* allele. TILLING? Mutant screening?”

A detailed description of this allele has been added (line 222).

“p.3 Would you write every “root-active CLEs” in the main text to make easy to read? (For example, root-active CLEs; CLE6, 8, 9/10, and 45)”

Yes, we now indicate them in detail in the text (line 169).

“p.6 The sentence ‘Root protophloem differentiation and thereby root growth of *bam3* loss-of-function mutants is not impaired by exogenously applied CLE45, suggesting that *BAM3* could act as a CLE45 receptor (Depuydt et al, 2013).’ is a strong expression. The *bam3* mutants reduced their primary root length by CLE45 but the reduction is quite milder than wild-type. This suggests there are *BAM3*-independent CLE45 perception pathways, which would explain why the *bam3* mutant does not have any obvious phenotype in root development.

The *bam3* mutant is indeed very resistant to CLE45 by comparison. We believe this statement has to be seen in the context that *bam3* is resistant to CLE45 levels that completely suppress protophloem development in wild type, as stated in the text.

-We suggest that the authors change the labelling of Figure 1F? It could easily be simplified (e.g. using only one letter repeatedly instead of 2 numbers).”

Yes, indeed labeling was suboptimal; this has been corrected.

2nd Editorial Decision

19 April 2017

Thank you for the submission of your revised manuscript to EMBO reports. We have now received the full set of referee reports that is copied below.

As you will see, all referees are positive about the study. Referee 1 however is not convinced about the co-localization of CLV2 and CRN. Referee 3 indicates that a statistical analysis as well as a methodological description of the qRT-PCR results are missing. Please address these points in the revised version.

From the editorial side, there are also a few things that we need before we can proceed with the official acceptance of your study.

- It is a precondition for publication in EMBO reports that authors agree to make all data freely available, where possible in an appropriate public database. In the case of sequencing data these should be deposited in the ArrayExpress or GEO databases and the accession number provided in the manuscript.

- Your manuscript will be published as Scientific Report. In this case the Results and Discussion sections have to be combined. The main text (excluding materials and methods) should not exceed 25,000 {plus minus} 2,000 characters.
- Please reformat the references according to the numbered style of EMBO reports. The respective EndNote file can be downloaded from our Author Guidelines, if required.
- The labels in Fig. 5A might be difficult to read in the final print size (in particular the "a" and "b" labels). Please ensure to use a font size of at least 7-8 pt at final size in all figures.
- Please add information about sample size, the nature of the bars (SEM, SD) and the test to calculate the p-values to the legends of Fig. EV1A, EV2C, and EV2D. Please also add information about the size of the scale bar to the legend of Fig. EV4.
- Please supply an ORCID ID for the co-corresponding author Prof. Hothorn, since all corresponding authors are required to supply an ORCID ID for their name.
- Please provide each EV table as separate file and remove the EV table legends from the main manuscript.
- Table EV3 seems to be never mentioned in the text. Please include a reference to this table where appropriate.
- Please submit Figures 5 and EV4 in portrait orientation.

Referee #1:

The revised manuscript has been improved substantially, and this reviewer commend the authors for evaluating the bioactivity of N-term tyrosinated CLE peptides as my suggestion and identifying that they lost bioactivities (and resulted in low affinity binding). It is also commending that the authors performed a whole-genome sequencing to find out that *serk1-1* genome harbors an additional mutation in *BAM3*. The presence of background mutations has been causing mis-interpretation of gene functions as well as phenotypic interpretation, and the effort by the authors set a high standard for this type of study. Overall the revised manuscript clarifies the role of CRN for CLE45-BAM3 signaling and implies a yet another complexity in CLE-peptide signaling at the level of SERK/BAK co-receptor specificity.

Some specific comments for publication:

- 1) NSG sequencing data: I do not find the accession number for the *serk1-1* WGS. Please submit the short read data and supply the accession number accordingly.
- 2) Relationship and colocalization of CLV2 and CRN: The data in the revised manuscript do not clearly support the relationship and co-localization of CLV2 and CRN. For instance, in the new figures EV4 B-D, I can find only minor co-localization between CLV2 or CRN with ER marker. For example, CRN-mTFP1 (EV4C) looks similar to BAM3-CITRINE (EV4A) which is localized to plasma membrane. In contrast, ER-mCHERRY (EV4C') is localized to dot-like structures. The overlay (EV4C') rather indicates the different subcellular localization between CRN (plasma membrane) and ER marker. Likewise, in both new Figures EV3A/A' (*clv2* background) and EV3F/F' (*crn* background), CLV2pro-CLV2-CITRINE seems mostly diffused in the cell. CRN-TurboRFP (Fig 4A, 4G) and CRN-mTFP1 (Fig EV4C) seem to show different localization patterns, which could be also due to the influence of internal CLV2. Please clarify these points.

Referee #2:

The authors have made every reasonable effort to improve the manuscript. With the new affinity data, an important question has been addressed.

Referee #3:

I agree that the authors' responses and new data they provided are all relevant and sufficient. I only have some minor points:

- Can they perform statistical analysis for the qPCR results? If they did, please write the details in the text or legend.
- Please add the method of qRT-PCR in the method section and SEM or SD in legend.

2nd Revision - authors' response

21 April 2017

Thank you for the positive evaluation of our manuscript. We now submit a revised version that addresses the remaining editorial and reviewer comments; please find a point-by-point response below.

Editorial comments:

- It is a precondition for publication in EMBO reports that authors agree to make all data freely available, where possible in an appropriate public database. In the case of sequencing data these should be deposited in the ArrayExpress or GEO databases and the accession number provided in the manuscript.

Yes, please excuse our oversight; we have now submitted the sequencing data at the Short Read Archive, with details such as accession numbers provided in the Methods (line 408). The data have already been released.

- Your manuscript will be published as Scientific Report. In this case the Results and Discussion sections have to be combined. The main text (excluding materials and methods) should not exceed 25,000 {plus minus} 2,000 characters.

As discussed per email, our paper was originally formatted as an EMBO Reports article, we apologize for not selecting the right option in the submission system.

- Please reformat the references according to the numbered style of EMBO reports. The respective EndNote file can be downloaded from our Author Guidelines, if required.

We have now reformatted the references using the Endnote style provided on your web site.

- The labels in Fig. 5A might be difficult to read in the final print size (in particular the "a" and "b" labels). Please ensure to use a font size of at least 7-8 pt at final size in all figures.

Yes, these are superscript, and we have now increased font size to make sure the labels will be legible.

- Please add information about sample size, the nature of the bars (SEM, SD) and the test to calculate the p-values to the legends of Fig. EV1A, EV2C, and EV2D. Please also add information about the size of the scale bar to the legend of Fig. EV4.

We apologize for this oversight; the details have now been added (lines 715, 736 and 782).

- Please supply an ORCID ID for the co-corresponding author Prof. Hothorn, since all corresponding authors are required to supply an ORCID ID for their name.

We now also provide the OrcID for Prof. Hothorn (0000-0002-3597-5698).

- Please provide each EV table as separate file and remove the EV table legends from the main manuscript.

The EV Tables are now provided separately; the legends have been removed.

- Table EV3 seems to be never mentioned in the text. Please include a reference to this table where appropriate.

Thank you for pointing this out, we now refer to Table EV3 in the Methods (line 428).

- Please submit Figures 5 and EV4 in portrait orientation.

All figures are now in portrait orientation.

Referee #1:

1) NSG sequencing data: I do not find the accession number for the *serk1-1* WGS. Please submit the short read data and supply the accession number accordingly.

Yes, please excuse our oversight; we have now submitted the sequencing data at the Short Read Archive, with details such as accession numbers provided in the Methods (line 408).

2) Relationship and colocalization of CLV2 and CRN: The data in the revised manuscript do not clearly support the relationship and co-localization of CLV2 and CRN. For instance, in the new figures EV4 B-D, I can find only minor colocalization between CLV2 or CRN with ER marker. For example, CRN-mTFP1 (EV4C) looks similar to BAM3-CITRINE (EV4A), which is localized to plasma membrane. In contrast, ER-mCHERRY (EV4C') is localized to dot-like structures. The overlay (EV4C') rather indicates the different subcellular localization between CRN (plasma membrane) and ER marker. Likewise, in both new Figures EV3A/A' (*clv2* background) and EV3F/F' (*crn* background), CLV2pro-CLV2-CITRINE seems mostly diffused in the cell. CRN-TurboRFP (Fig 4A, 4G) and CRN-mTFP1 (Fig EV4C) seem to show different localization patterns, which could be also due to the influence of internal CLV2.

We agree with the reviewer that these co-localizations are not absolute, in line what has been published, and we did not mean our statements to be interpreted this way. To clarify this, we have now changed our wording ("CLV2 and CRN fusion proteins were mostly found inside cells and co-localized substantially with an endoplasmic reticulum marker", line 240). We also agree that in the tobacco transformations, endogenous CLV2/CRN might influence localization, we now point this out as well ("Some plasma membrane localization could be observed at variable degrees in replicate experiments, which might be due to endogenous CLV2/CRN proteins, because as previously reported [54], co-expression of CLV2 and CRN resulted in increased delivery of both fusion proteins to the plasma membrane", line 242).

Referee #3:

- Can they perform statistical analysis for the qPCR results? If they did, please write the details in the text or legend.

- Please add the method of qRT-PCR in the method section and SEM or SD in legend.

We apologize for omitting this information; we have now added the statistics details to the figure legend (line 694) and experimental details to the Methods section (line 429).

3rd Editorial Decision

04 May 2017

Thank you for your patience while we have gone through your revised manuscript. There are now only a few minor issues that need to be corrected and I am therefore writing with an 'accept in principle' decision, which means that I will be happy to accept your manuscript for publication once these corrections have been addressed, as follows.

- We had earlier asked to supply Figs 5 and EV4 in portrait format. I noticed that they appear rather square now. Could you please provide these figures in a true portrait form and rearrange the figure panels if necessary? This will ensure that they can be correctly displayed during typesetting of the article and in the print version.

- Please supply the EV tables either in .doc or .excel format. I apologize for not having stated this explicitly in my last letter, when we asked you to provide a separate file for each table. If you upload the tables in .xls format, the legend can be given in the first row of the table.

If all remaining corrections have been attended to, you will then receive an official decision letter from the journal accepting your manuscript for publication in the next available issue of EMBO reports. This letter will also include details of the further steps you need to take for the prompt inclusion of your manuscript in our next available issue.

3rd Revision - authors' response

04 May 2017

The authors made the requested changes and submitted a final version of their manuscript and its accompanying files.

4th Editorial Decision

05 May 2017

I am very pleased to accept your manuscript for publication in the next available issue of EMBO reports. Thank you for your contribution to our journal.

YOU MUST COMPLETE ALL CELLS WITH A PINK BACKGROUND

Corresponding Author Name: C. Hardtke & M. Hothorn
Journal Submitted to: EMBO Reports
Manuscript Number: EMBOR-2016-435352